# Three-color single molecule imaging shows WASP detachment from Arp2/3 complex triggers actin filament branch formation

Benjamin A Smith[1][†][‡], Shae B Padrick[2][†], Lynda K Doolittle[2], Karen Daugherty-Clarke[3,4], Ivan R Corrêa Jr[5], Ming-Qun Xu[5], Bruce L Goode[3,4], Michael K Rosen[2]*, Jeff Gelles[1]*

[1]Department of Biochemistry, Brandeis University, Waltham, United States; [2]Department of Biophysics, Howard Hughes Medical Institute, University of Texas Southwestern Medical Center, Dallas, United States; [3]Rosenstiel Basic Medical Sciences Research Center, Brandeis University, Waltham, United States; [4]Department of Biology, Brandeis University, Waltham, United States; [5]New England Biolabs, Ipswich, United States

**Abstract** During cell locomotion and endocytosis, membrane-tethered WASP proteins stimulate actin filament nucleation by the Arp2/3 complex. This process generates highly branched arrays of filaments that grow toward the membrane to which they are tethered, a conflict that seemingly would restrict filament growth. Using three-color single-molecule imaging in vitro we revealed how the dynamic associations of Arp2/3 complex with mother filament and WASP are temporally coordinated with initiation of daughter filament growth. We found that WASP proteins dissociated from filament-bound Arp2/3 complex prior to new filament growth. Further, mutations that accelerated release of WASP from filament-bound Arp2/3 complex proportionally accelerated branch formation. These data suggest that while WASP promotes formation of pre-nucleation complexes, filament growth cannot occur until it is triggered by WASP release. This provides a mechanism by which membrane-bound WASP proteins can stimulate network growth without restraining it.

*For correspondence: michael. rosen@utsouthwestern.edu (MKR); gelles@brandeis.edu (JG)

[†]These authors contributed equally to this work

[‡]Present address: Biogen Idec, Cambridge, United States

## Introduction

Control of actin dynamics is essential to many cellular processes, including motility, vesicle trafficking, and cell division (*Pollard and Cooper, 2009*). The Actin related protein 2/Actin related protein 3 (Arp2/3) complex nucleates new (*daughter*) filaments from the sides of existing (*mother*) filaments in response to activating stimulus from the Wiskott-Aldrich Syndrome Protein (WASP) family (*Pollard, 2007*; *Padrick and Rosen, 2010*; *Campellone and Welch, 2010*). Membrane-associated WASP proteins integrate upstream signals and activate Arp2/3 complex in the correct place and time to produce actin structures that perform a variety of cellular functions including motility. The verprolin homology-central-acidic (VCA) domain of WASP family proteins binds to monomeric actin and the Arp2/3 complex, and is both necessary and sufficient for the WASP proteins to stimulate nucleation (*Machesky and Insall, 1998*; *Miki and Takenawa, 1998*; *Machesky et al., 1999*; *Rohatgi et al., 1999*; *Pollard, 2007*).

VCA acts to promote daughter nucleation by Arp2/3 complex in several ways. VCA engagement promotes a conformational change in Arp2/3 complex that repositions Arp2 and Arp3 (*Robinson et al., 2001*; *Goley et al., 2004*; *Rodal et al., 2005*; *Rouiller et al., 2008*; *Xu et al., 2012*; *Hetrick et al., 2013*). This conformational change is thought to be required to initiate daughter filament growth.

**eLife digest** Most cells are neither perfect spheres nor amorphous blobs, but instead have characteristic shapes that enable them to carry out specific roles within tissues or organs. These shapes are established by a type of scaffolding, called the cytoskeleton, that gives structure to the cell, and also forms networks over which other proteins, and even organelles, can travel.

The filaments that make up the cytoskeleton are composed of various proteins, one of which is called actin. Cellular actin filaments can grow by adding new actin molecules, and actin filaments can also have 'branches' that fork out from the mother filament. Branches grow out of an assembly of seven proteins known as the Arp2/3 complex, which attaches to the side of the mother filament. Branch growth is triggered by binding to the Arp2/3 complex of an additional protein, WASP, but the sequence of events required to initiate a new branch is not well understood. In particular, WASP is bound to cell membranes; at some point it must detach from the Arp2/3 complex so that the nearness of the membrane does not interfere with the growth of branches. Now, Smith et al. uncover how branch formation is triggered, and define a new role played by WASP in this process.

It is known that a specific region of the WASP protein called the VCA domain binds to both the Arp2/3 complex and actin. Smith et al. studied how this domain could initiate branch formation, and showed that a pair of VCA domains linked to each other, along with an Arp2/3 complex, could interact jointly with an existing actin filament before a new branch formed. However, new branches did not form unless the VCA-domain pair detached from the actin filament, leaving the Arp2/3 complex behind. Additionally, Smith et al. found that mutant VCA-domain pairs detached from the actin filament at different rates, which then determined the chance that a new branch formed.

These findings—and those of Helgeson and Nolen published concurrently in *eLife*—suggest that, in cells, two WASP proteins first recruit the Arp2/3 complex to the membrane, and that together they interact with an existing actin filament. The WASP proteins then release the filament, and only then does the Arp2/3 complex initiate the formation of an actin branch. Since the Arp2/3 complex is no longer attached to WASP, subsequent growth of the branch is not physically limited by linkage to the membrane.

Further, the initial monomers of the nucleated filament are delivered by the V region, or WASP homology 2 (WH2) domain (*Rohatgi et al., 1999*; *Hertzog et al., 2004*; *Irobi et al., 2004*; *Chereau et al., 2005*; *Padrick et al., 2011*). However, the Arp2/3-VCA-actin complex is not active on its own; an additional stimulus must be provided by the mother filament (*Mullins et al., 1998*; *Achard et al., 2010*). Thus, when stimulated by VCA the Arp2/3 complex only nucleates filaments from the sides of existing filaments. This produces branched filament arrays in vitro (*Mullins et al., 1998*; *Machesky et al., 1999*; *Blanchoin et al., 2000*; *Amann and Pollard, 2001*; *Achard et al., 2010*), which resemble the branched networks found in cells (*Svitkina and Borisy, 1999*; *Vinzenz et al., 2012*). An additional feature of the system is that simultaneous binding of two VCA peptides to Arp2/3 complex greatly potentiates daughter nucleation (*Padrick et al., 2011*). It is likely that WASP oligomerization is a broadly used mechanism of activation (*Padrick et al., 2008*), and a number of cellular factors that dimerize or multimerize WASP have been identified (*Padrick and Rosen, 2010*).

The molecular interactions and structural rearrangements outlined above contribute to VCA acting at more than one step in the nucleation pathway, although the pathway is not fully defined. It is established that VCA stimulates branch formation by accelerating the association of Arp2/3 complex with the mother filament (*Smith et al., 2013*). In addition, there is evidence from kinetic analyses that a VCA-dependent 'activation step' follows filament binding during nucleation (*Marchand et al., 2001*; *Zalevsky et al., 2001*; *Beltzner and Pollard, 2008*; *Smith et al., 2013*). The nature of this step remains unknown but has been hypothesized to arise from conformational changes in the Arp2/3 complex (*Marchand et al., 2001*; *Beltzner and Pollard, 2008*). The activation step substantially limits the efficiency of nucleation (*Smith et al., 2013*).

An interesting feature of WASP activation of daughter nucleation is that Arp2/3 complex must associate with membrane bound activators at an early stage in the process and yet be separated from those activators at a subsequent stage. In cells, branched filament networks have their barbed ends directed toward membranes (*Small et al., 1978*; *Svitkina and Borisy, 1999*; *Pollard and Borisy, 2003*;

*Vinzenz et al., 2012*). The characteristic geometry of the branches nucleated by the Arp2/3 complex dictates that both mother and daughter filaments grow toward the membrane. However, WASP proteins are linked with activators on the membrane (*Padrick and Rosen, 2010*), so that VCA-bound Arp2/3 complex should be tethered to the membrane. This tethering creates a steric problem, in that the growing ends of the filaments are held against, and possibly have their growth limited by, the membrane. However, this problem is eventually resolved (*Figure 1*, right). In lamellipodia, VCA-containing WAVE proteins stay associated with the leading edge, while the Arp2/3 complex is distributed throughout the actin mesh (*Lai et al., 2008*). In both budding and fission yeasts, Arp2/3 complex is separated from membrane-bound activators during endocytosis (*Kaksonen et al., 2003*; *Sirotkin et al., 2005*). Analogously, in propulsive actin 'comet tails' Arp2/3 complex is found throughout the tail while its activators stay (largely) associated with the motile bacterium, virus, or vesicle (*Welch et al., 1997*; *Egile et al., 1999*; *Loisel et al., 1999*; *Taunton et al., 2000*; *Weisswange et al., 2009*). In vitro, branches are released from the budding yeast WASP family member Las17 (*Martin et al., 2006*). In all of these systems Arp2/3 complex can disengage from the surface attached activator within a short time of the onset of daughter filament growth.

These observations raise a fundamental question: how is the binding of Arp2/3 complex to filament sides coordinated with binding and release of VCA and the initiation and growth of the daughter filament (*Figure 1*)? Previous studies have proposed that VCA may dissociate from Arp2/3 complex prior to initiation of daughter filament growth. The observation that ATP hydrolysis by Arp2 decreases affinity for VCA suggested a possible trigger for dissociation (*Dayel et al., 2001*; *Dayel and Mullins, 2004*), although hydrolysis was later shown to be dispensable for filament nucleation and to instead control disassembly of the branch (*Martin et al., 2006*; *Ingerman et al., 2013*). Consideration of biochemical and structural data on WH2-actin interactions (*Egile et al., 1999*; *Higgs et al., 1999*; *Hertzog et al., 2004*; *Irobi et al., 2004*; *Chereau et al., 2005*; *Boczkowska et al., 2008*) led to speculation that during Arp2/3-mediated nucleation, the WH2 region of VCA may initially block the barbed end of actin delivered to the Arp2 or Arp3 subunit and thus must move or dissociate prior to daughter filament elongation (*Padrick et al., 2011*; *Ti et al., 2011*; *Hetrick et al., 2013*). However, direct observation of the sequence of molecular events associated with the initiation of daughter filament growth and the release of VCA has been lacking.

Multi-wavelength single-molecule fluorescence colocalization methods (*Hoskins et al., 2011*; *Friedman and Gelles, 2012*) are a powerful approach to elucidating the reaction pathways and identifying key regulated steps in processes that involve multiple macromolecular components. Here

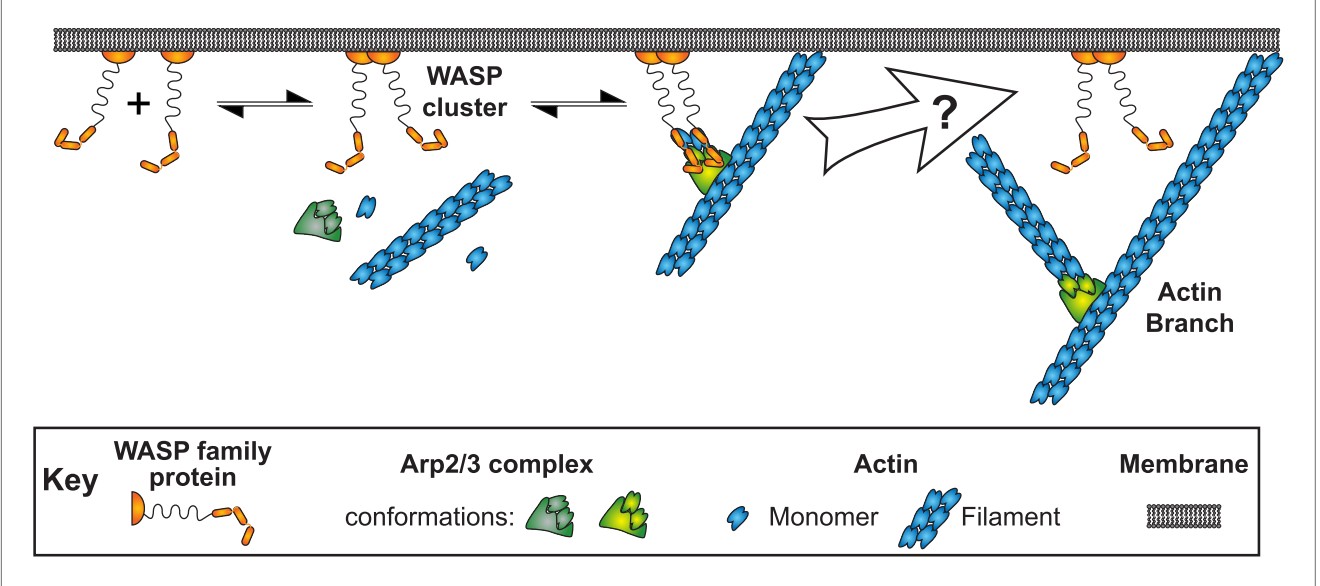

**Figure 1**. Pathway of Arp2/3 complex mediated actin branch formation activated by WASP protein dimers on the inside surface of a cell membrane, as deduced from previous studies. Within the white arrow Arp2/3 complex is activated by VCA, detaches from the membrane and initiates daughter filament elongation. The order of these steps and how they are coordinated remains unclear.

we used three-color single molecule fluorescence experiments to directly visualize the sequence and rates of the key steps in the pathway through which VCA dimers, Arp2/3 complexes, and actin filaments associate with one another and generate a new actin branch. The work reveals that the activation step in nucleation is likely the release of VCA dimers from the nascent branch, such that VCA dissociation is the trigger for daughter filament growth. The problem of filament growth against a membrane to which it is tethered is therefore solved by a mechanism in which release from the membrane tether is required for filament initiation.

## Results

### VCA is released from filament-bound Arp2/3 complex before initiation of daughter filament growth

To follow the coordination of VCA association with Arp2/3 complex during actin branch formation we labeled each protein with a fluorescent probe and visualized their colocalization dynamics using colocalization single molecule spectroscopy (CoSMoS) (*Friedman et al., 2006*; *Hoskins et al., 2011*; *Friedman and Gelles, 2012*). Actin was labeled with a blue-excited dye (on 10% of monomers) and tagged with biotin (on 1% of monomers) to enable tethering to microscope slides. *Saccharomyces cerevisiae* Arp2/3 complex was labeled with a red-excited dye targeted to a SNAP tag fused to the C-terminus of the Arc18 (ArpC3) subunit (*Smith et al., 2013*). We used a green-excited, Cy3 dye bis-maleimide derivative to label and covalently dimerize the VCA from N-WASP, hereafter called diVCA, which includes the second WH2 motif (V) through the C-terminus of the protein. Like other dimeric VCAs (*Padrick et al., 2008*) (*Figure 2—figure supplement 1*), this diVCA construct was able to stimulate the activity of the Arp2/3 complex (*Figure 2—figure supplement 2*; 'Materials and methods') at low-nanomolar concentrations suitable for single molecule imaging, whereas labeled monomeric VCA constructs did not.

Using this combination of tagged proteins (*Figure 2A*) we directly observed individual Arp2/3 complex and diVCA molecules binding to immobilized actin filaments and nucleating new branches. During this process, single molecules of Arp2/3 complex and diVCA were observed to bind together to locations on filament sides (e.g., *Figure 2B–C* at $t = 0$; *Figure 2—figure supplement 3*). In nearly all cases (83 ± 9% S.E., based on 877 Arp2/3 complex observations), Arp2/3 complex and diVCA arrived simultaneously within the experimental time resolution (0.15 s), indicating that diVCA was bound to Arp2/3 complex prior to filament engagement, and that both proteins bound to filaments as a unit ('Materials and methods'). The sparse labeling of actin and high fluorescence intensity of mother filaments did not permit us to detect the arrival of actin monomers with VCA and Arp2/3 complex. However, under the reaction conditions (5 nM diVCA and 1 μM actin), with a $K_D$ of diVCA for actin of ~300 nM (which is only slightly altered by the presence of Arp2/3 complex, *Figure 2—figure supplement 4*), we expect ~77% of diVCAs to have at least one actin bound. Thus, most of the diVCA-Arp2/3 complexes observed to bind mother filament should contain actin and thus have all the molecular factors needed for nucleation. We hereafter refer to this filament-bound complex as the 'nascent branch', an intermediate in the pathway to daughter filament assembly.

Most nascent branches, although containing the necessary components for nucleation, dissociated quickly (typically in <1 s) without producing a daughter filament. This is consistent with our previous observations that the vast majority of Arp2/3 complex binding events are non-productive (*Smith et al., 2013*). In nearly all cases (97.3 ± 0.4%, based on 758 nascent branch observations), diVCA and Arp2/3 complex dissociated simultaneously as a unit (*Figure 2—figure supplement 3*). The on-filament lifetime distributions of diVCA and Arp2/3 complex have identical short ($\tau_1$ and $\tau_2$) components (*Figure 2D* at time <1 s), consistent with the conclusion that the two molecules are released from the filament as an Arp2/3-diVCA complex.

In contrast to the large majority of non-productive filament encounters, a small fraction of Arp2/3-diVCA filament binding events led to formation of a daughter filament. A merit of the single-molecule approach is that we could characterize these rare productive events (e.g., *Figure 2B–C*) independently of the excess of non-productive events. In the productive events, Arp2/3 complex and diVCA release were not simultaneous. There was no evidence that the Arp2/3 complexes which formed branches ever dissociated; the value of $\tau_3$ is set by the photobleaching lifetime of the dye-labeled Arp2/3 complex (*Smith et al., 2013*). In contrast, diVCA dissociated rapidly (typically in <1 s) from the productive Arp2/3-diVCA-filament complexes. Consistent with this observation, the on-filament lifetime distribution of diVCA lacks a long component ($\tau_3$) that is present in the on-filament lifetime

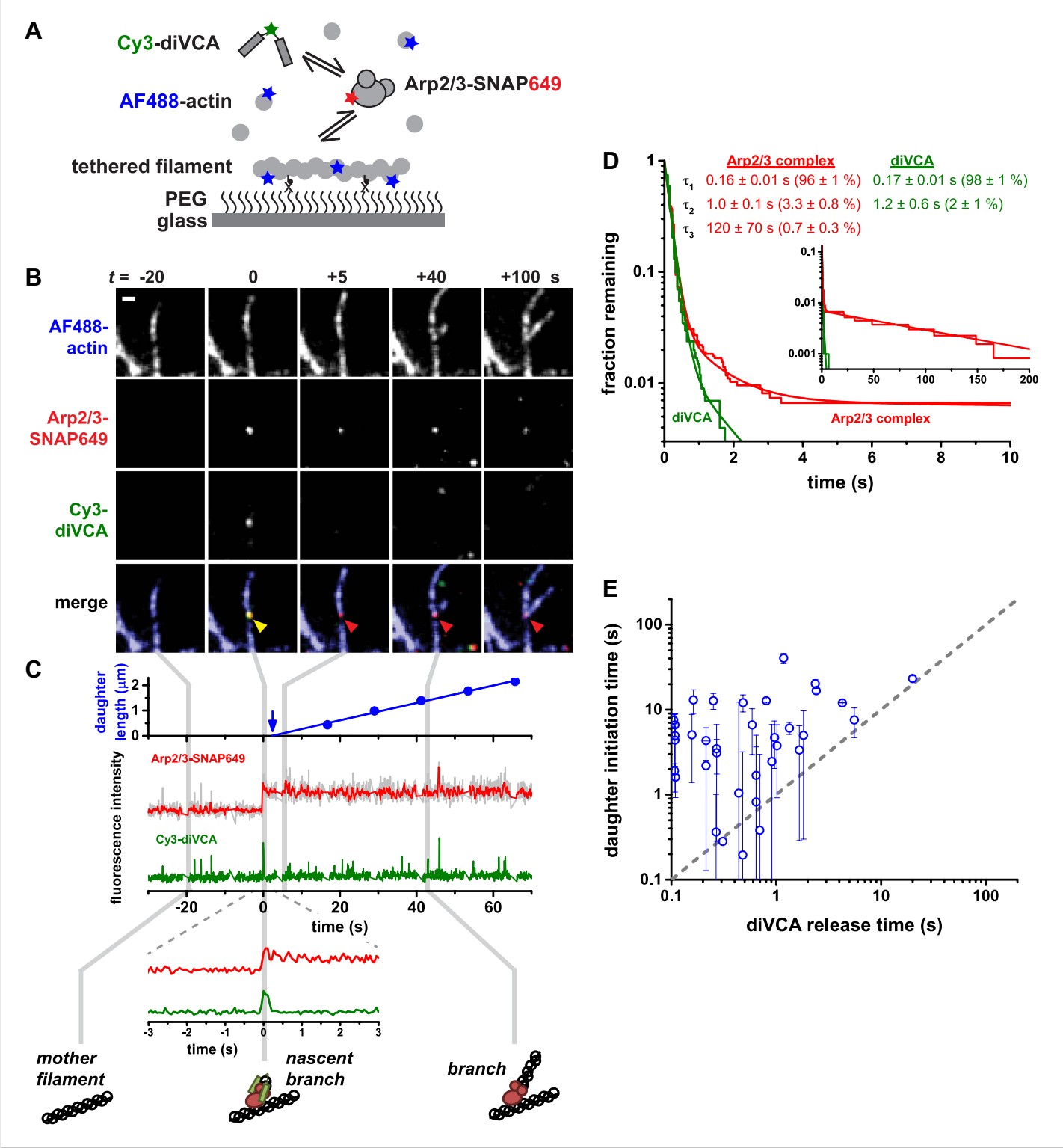

**Figure 2**. Rapid release of dimeric VCA from the nascent branch precedes nucleation. (**A**) Design of an experiment to observe diVCA-activated branch nucleation by Arp2/3 complex on the sides of surface-immobilized actin filaments. Blue, green and red stars denote fluorescent dye labels AlexaFluor 488 (AF488), Cyanine 3 (Cy3), and Dy649 that are excited with blue, green, and red lasers, respectively. (**B**) Image sequence of the same microscope field of view taken at each of the three dye wavelengths (rows) at five selected time points (*t*; columns). Images record the colocalization of an individual Arp2/3 complex and diVCA molecule at *t* = 0 (yellow arrowhead) followed by nucleation and growth of a daughter filament at that location (red arrowhead).
*Figure 2. Continued on next page*

*Figure 2. Continued*

Solution contained 5 nM Cy3-diVCA, 5 nM SNAP-tagged Arp2/3 complex labeled with Dy649 (Arp2/3-SNAP649), and 1 µM actin, 10% AF488-labeled. Bar: 1 µm. See **Video 1**. (**C**) Recordings of daughter filament length and branch site fluorescence intensities from the nucleation event in **B**. Arrow marks the time of daughter filament nucleation estimated by extrapolating the daughter length fit line to zero length (**Smith et al., 2013**). Plot at bottom is a magnified view showing that Arp2/3 complex and diVCA labels appear simultaneously ($t = 0$) followed by rapid release of diVCA ($t = 0.2$ s). (**D**) Cumulative lifetime distributions of Arp2/3 complex and diVCA on filament sides after binding of an Arp2/3-diVCA complex to the filament ($N = 752$). Smooth lines indicate two- (diVCA) or three-exponential (Arp2/3 complex) fits yielding the indicated fit parameters ('Materials and methods'). Main plot shows the data for time <10 s; inset shows the full distribution with the exception of one outlier. (**E**) Comparison of the time (±S.E.) of daughter filament initiation with the time of diVCA release from the nascent branch in individual branch nucleation events by diVCA-Arp2/3 complexes.

The following figure supplements are available for figure 2:

**Figure supplement 1**. N-WASP VCA dimers bind tightly to Arp2/3 complex and stimulate its actin nucleation activity.

**Figure supplement 2**. VCA dimers crosslinked through Cy3 stimulate Arp2/3 to a similar extent as GST-VCA dimers.

**Figure supplement 3**. Arp2/3 complex and diVCA usually bind to and release from filaments as a unit when no daughter filament is formed.

**Figure supplement 4**. Association with Arp2/3 complex does not affect binding of VCA to actin.

distribution of Arp2/3 complex (**Figure 2D** at time > approximately 4 s). Thus, the data demonstrate that daughter nucleation is essentially always accompanied by Arp2/3 complex retention and diVCA release.

To determine whether diVCA release occurs before or after the onset of daughter filament growth, we measured the time at which each daughter filament initiated elongation by extrapolating daughter length records (as in **Figure 2C**, top). These filament initiation times were then compared to the times of diVCA release from the same nascent branch. Filament initiation time measurements were imprecise because of the uncertainties inherent in measuring daughter filaments of sub-micrometer lengths. Nevertheless, within this experimental uncertainty we observed that the initiation of daughter filament growth always occurred at or after the time of diVCA release (41 observations; **Figure 2E**). This was true even in the comparatively rare cases in which diVCA persisted on the nascent branch for times >1 s before dissociating. Taken together, these data suggest that the daughter filament cannot initiate unless and until VCA is released from the nascent branch. Thus, diVCA release may serve as the trigger for daughter growth.

## VCA does not bind Arp2/3 complex after branch formation

Next we asked whether diVCA can bind to Arp2/3 complex after branches have formed, in order to better understand how WASP recruits and activates free Arp2/3 complex yet does not stay bound to Arp2/3 complex in branch junctions and restrict network growth ('Introduction'). To address this question, we tethered individual dye- and biotin-labeled Arp2/3 complexes to the microscope slide and visualized the binding of freely diffusing diVCA and (non-biotinylated) actin filaments (**Figure 3A**). Most of the tethered

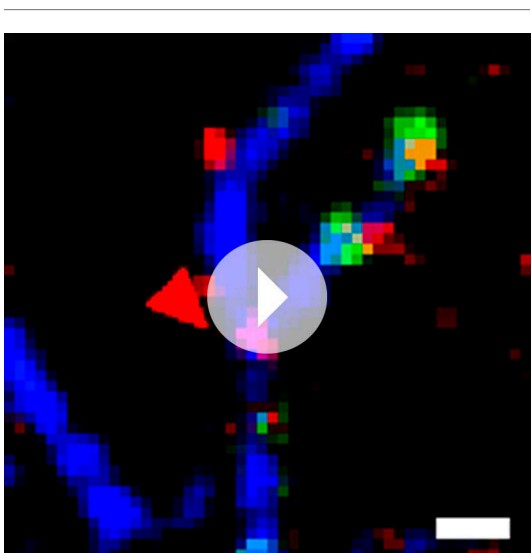

**Video 1**. The diVCA-stimulated actin branch formation event shown in **Figure 2B–C**. Red: Arp2/3-SNAP649 (5 nM in solution); green: Cy3-diVCA (5 nM); blue: actin-AF488 (1 µM, 10% labeled). Arp2/3 complex and diVCA images were recorded every 0.05 s; actin images were recorded every ~12 s. Playback rate: real time. Bar: 1 µm. Many diVCA-Arp2/3 complexes are observed to transiently associate with actin filaments. One such nascent branch complex (yellow arrowhead) releases diVCA shortly after it appears, leaving Arp2/3 complex stably associated with the mother filament (red arrowhead), where it subsequently initiates daughter filament elongation.

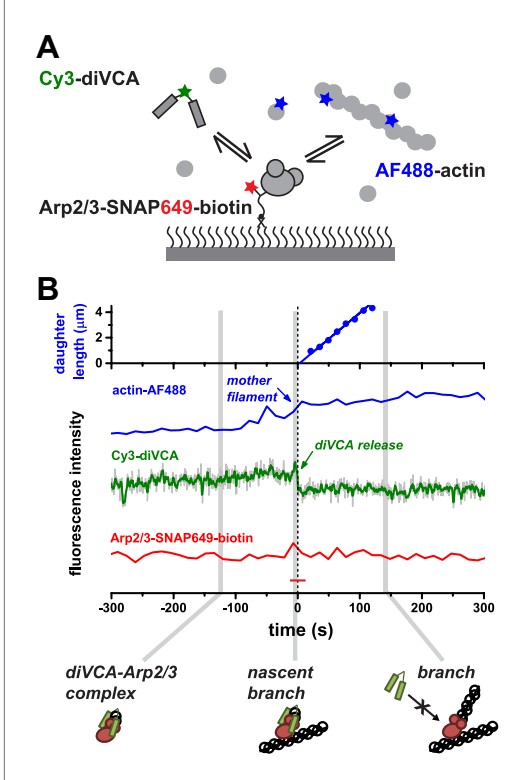

**Figure 3**. VCA dimers form long-lived complexes with Arp2/3 complex before filament binding but not after branch formation. (**A**) Experimental design to observe diVCA binding and nucleation of actin filaments on immobilized Arp2/3 complexes. Arp2/3-SNAP was tethered to the slide surface via a bi-functional SNAP substrate that incorporated both a Dy649 dye and a biotin-terminated PEG chain; we monitored binding of fluorescently labeled diVCA and actin filaments from solution. (**B**) Example record showing the length of a nucleated daughter filament and the fluorescence intensity from actin (blue) and diVCA (green) at an individual tethered Arp2/3 complex molecule. The solution contained 1 μM actin (10% AF488 labeled) and Cy3-diVCA (2 nM). Fluorescence from the tethered Arp2/3 complex (red trace) remained steady and above background (red dash) throughout. Schematics show the inferred complexes present at the indicated times. Time zero is the time of diVCA release.

Arp2/3 complexes (>80%) were observed to bind diVCA. Binding lasted for tens or hundreds of seconds when no filaments were nearby (e.g., *Figure 3B* at time <0). With 2 nM diVCA in solution, individual tethered Arp2/3 complexes were nearly continuously occupied, suggesting a dissociation equilibrium constant $K_D$ < 2 nM similar to bulk affinity measurements on other diVCA and Arp2/3 complex species (*Figure 2—figure supplement 1E*). Further, we never observed the nucleation of a new actin filament from an isolated surface-tethered diVCA-Arp2/3 complex, consistent with previous conclusions that Arp2/3 complex cannot nucleate a daughter unless it is bound to a pre-existing mother filament (*Machesky et al., 1999*; *Blanchoin et al., 2000*; *Achard et al., 2010*).

In some cases, we did observe association of a mother filament (formed in solution) with some tethered Arp2/3-diVCA complexes, followed by growth of a daughter filament. This yielded a branch junction that remained stably colocalized with the tethered Arp2/3 complex. In those events (e.g., *Figure 3B* at time >0), branch formation essentially abolished binding of diVCA; any appearance of diVCA on branch junctions was transient (<0.1 s) and occurred at a low frequency (~0.7 × 10^6 M^−1 s^−1; N = 2) comparable to non-specific binding at randomly chosen points on the microscope slide (1.0 ± 0.6 × 10^6 M^−1 s^−1). The data demonstrate that the affinity of diVCA for isolated Arp2/3 complex is high, whereas the diVCA affinity for Arp2/3 complex in the branch junction is comparatively low. The low affinity of diVCA for the branch is consistent with data from previous studies (*Egile et al., 2005*; *Martin et al., 2006*). In addition, our measurements suggest that in the cell, once WASP proteins dissociate and the branch forms the Arp2/3 complex incorporated in the branch junction is unlikely to reassociate with membrane-linked WASP proteins and thus will not restrict filament network growth.

## Targeted mutations alter the kinetic stability of ternary complexes of diVCA, actin monomers, and Arp2/3 complex

To challenge the model that diVCA release from the nascent branch is required to initiate daughter filament growth, we next engineered a series of mutations in diVCA (*Figure 4A*; *Figure 4—figure supplement 1*). The goal was to modestly perturb VCA interactions with Arp2/3 complex or actin without altering the reaction pathway by which diVCA stimulates branch formation. Guided by previous biochemical data (*Zalevsky et al., 2001*; *Panchal et al., 2003*; *Chereau et al., 2005*), we mutated each of the three regions of N-WASP VCA. The D435S/A436D mutation in the V-region (diVCA-V*) was designed to perturb actin affinity (*Chereau et al., 2005*), whereas mutations in the C-region (I467A, diVCA-C*) and A-region (Δ486–488, diVCA-A*) were designed to perturb interactions with Arp2/3 complex (*Zalevsky et al., 2001*; *Panchal et al., 2003*).

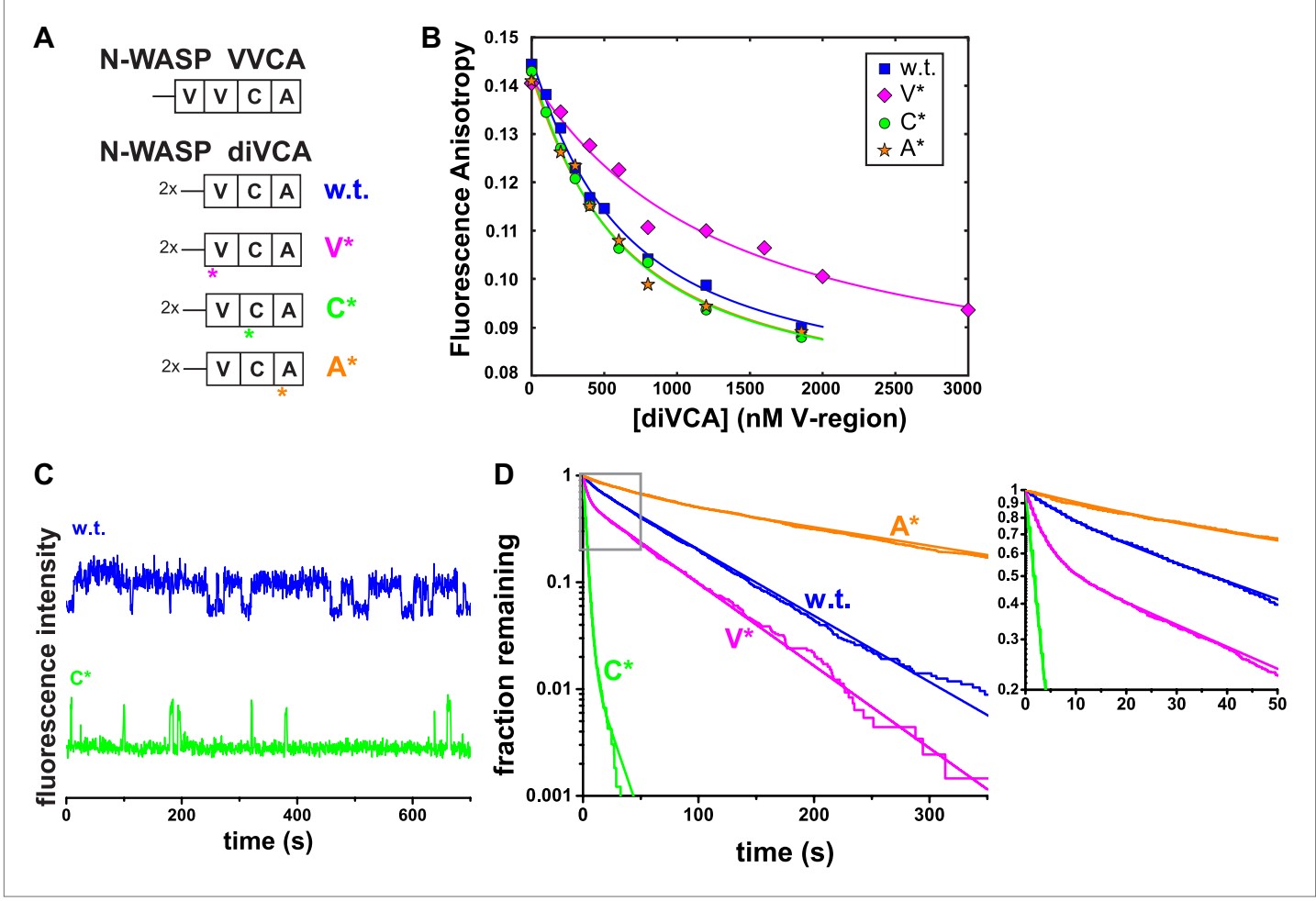

Figure 4. diVCA mutations alter the stability of Arp2/3 complex-diVCA-actin monomer assemblies. (A) Arrangement of V, C, and A domains in native N-WASP and in the diVCA constructs used in this study (w.t. is wild-type). Asterisks mark the domains bearing targeted mutations (substitution of one or two residues, or a three-residue deletion; *Figure 4—figure supplement 1A*) in the three mutant constructs. (B) Fluorescence anisotropy detected binding of AF488-labeled N-WASP VCA with rabbit muscle actin, in the presence of competitor wild-type (same data as in *Figure 2—figure supplement 4B*) or mutant Cy3-diVCA constructs (symbols). Data were fit (lines) with competition binding isotherms incorporating the coupled equilibria ('Materials and methods') yielding $K_D$ values 340 ± 60 (S.E.) nM for wild-type diVCA, 660 ± 80 nM for diVCA-V*, 260 ± 40 nM for diVCA-C*, and 250 ± 40 nM for diVCA-A*. (C) Example Cy3-diVCA fluorescence intensity records recorded on individual tethered Arp2/3 complexes (*Figure 3*): Cy3-diVCA wild-type or C* mutant (0.5 nM) molecules binding and dissociating in the presence of 1 µM actin monomers but no filament. (D) Cumulative lifetime distributions of diVCA-Arp2/3 complexes in the presence of monomeric actin observed in records like those in B. Smooth lines are biexponential fits (*Table 1*). Inset is a magnified view of the indicated data range.

The following figure supplements are available for figure 4:

Figure supplement 1. Design of the diVCA mutant constructs and characterization of Arp2/3 complex binding by the diVCA-C* mutant.

Figure supplement 2. Single molecule analysis of wild-type and mutant diVCA binding to and dissociating from tethered Arp2/3 complex.

We measured the affinity of the mutants for monomeric actin using a fluorescence anisotropy competition assay. Here, labeled VCA reported on binding to monomeric actin by an increase in fluorescence anisotropy (as in *Figure 2—figure supplement 4A*). As expected from the design, wild-type diVCA, diVCA-C*, and diVCA-A* had similar affinities for actin, while diVCA-V* bound more weakly (~650 nM vs ~300 nM for wild-type; *Figure 4B*).

We next compared the rates of wild-type and mutant diVCAs binding to and dissociating from Arp2/3 complex using surface-tethered Arp2/3 complexes in the presence of actin monomers but not bound to actin filament. For wild-type diVCA and all three mutants, we observed repeated association

and dissociation of the diVCA molecules with surface-tethered Arp2/3 complex (e.g., *Figure 4C*). These events allowed measurement of the lifetime distributions for the diVCA bound and dissociated states of Arp2/3 complex, and thus determination of the binding and dissociation rate constants. The mutations had only modest effects, at most 2.7-fold, on the rate of binding of diVCA to Arp2/3 complex (*Figure 4—figure supplement 2A–C*; *Table 1*). The dissociation rates varied over a wide range, with the A* mutant dissociating from Arp2/3 complex more slowly than wild-type, and the V* and C* mutants dissociating more rapidly (*Figure 4D*). Both wild-type and mutant complexes displayed lifetime distributions that were fit well with two exponential components (*Figure 4D*, *Figure 4—figure supplement 2D*, *Table 1*), indicating that at least two distinct diVCA-Arp2/3 complex assemblies or conformations were present. In a more detailed analysis with the V* mutant, we saw no evidence that the short and long lifetime components segregated into different subpopulations of individual Arp2/3 complexes. These observations suggest even individual diVCA-Arp2/3 complexes participated in multiple states. The presence of multiple different complexes is consistent with previous observations including that Arp2/3 complex has two (or more) binding sites for VCA (*Padrick et al., 2011*; *Ti et al.,*

**Table 1.** Colocalization kinetics and activities of diVCA and Arp2/3 complex

| | no VCA | diVCA w.t. | diVCA V* | diVCA C* | diVCA A* |
|---|---|---|---|---|---|
| Arp2/3 complex off filament (*Figure 4D*, *Figure 4—figure supplement 2*) | | | | | |
| $N$ (groups) | | 5 | 3 | 3 | 1 |
| $k_{V+}$ ($10^7$ M$^{-1}$ s$^{-1}$) | | 16±5 (SEM) | 7±3 (SEM) | 6±1 (SEM) | 16±1 (SE) |
| $N_V$ | | 1528 | 957 | 1339 | 1262 |
| $\tau_{V1}$ (s) | | 8±1 | 3.6±0.4 | 2.4±0.1 | 40±10 |
| $A_{V1}$ | | 24±3% | 42±2% | 98±2% | 28±9% |
| $\tau_{V2}$ (s) | | 61±3 | 56±3 | 14±6 | 250±30 |
| Arp2/3 complex on filament (*Figure 6B*) | | | | | |
| $N_A$ | 715 | 877 | 407 | 1089 | 597 |
| $k_{A+}$ ($10^4$ M$^{-1}$ s$^{-1}$) | 2.0±0.3 | 2.1±0.3 | 1.3±0.2 | 2±1 | 0.8±0.2 |
| $f_{AV}$ | | 0.83±0.09 | 0.7±0.1 | 0.59±0.08 | 0.7±0.1 |
| $f_{V-}$ | | 0.026±0.004 | 0.041±0.009 | 0.018±0.003 | 0.015±0.004 |
| $f_B$ | 0.006±0.002 | 0.008±0.002 | 0.029±0.008 | 0.015±0.003 | 0.013±0.003 |
| Arp2/3 complex at branch sites (*Figure 5A*, *Figure 6C–E*) | | | | | |
| $N_B$ | | 57 | 69 | 69 | 40 |
| $\tau_V^*$ (s) | | 0.7±0.1 | 0.54±0.08 | 0.37±0.04 | 0.7±0.2 |
| $k_V^*$ (s$^{-1}$) | | 0.04±0.01 | 0.08±0.02 | 0.05±0.01 | 0.022±0.008 |
| $k_B$ (M$^{-1}$ s$^{-1}$) | 120±40 | 160±50 | 320±90 | 200±100 | 100±30 |

Parameter descriptions:
$N$ = number of groups of observations used to calculate binding rate of diVCA to isolated Arp2/3 complexes.
$k_{V+}$ = second order rate constant for diVCA binding to Arp2/3 complexes.
$N_V$ = number of observations of diVCA on isolated Arp2/3 complexes.
$\tau_{V1}$ = first characteristic lifetime of diVCA on isolated Arp2/3 complexes.
$A_{V1}$ = percent of diVCA that dissociate from Arp2/3 complexes with time constant $\tau_{V1}$.
$\tau_{V2}$ = second characteristic lifetime of diVCA on isolated Arp2/3 complexes.
$N_A$ = number of observations of Arp2/3 complexes on the sides of select filaments.
$k_{A+}$ = second order rate constant for Arp2/3 complex binding filament sides (per filament subunit).
$f_{AV}$ = fraction of Arp2/3 complexes that bind filament sides coincident with diVCA.
$f_{V-}$ = fraction of diVCA-Arp2/3-filament complexes that release diVCA.
$f_B$ = fraction of diVCA-Arp2/3-filament complexes that nucleate a daughter filament.
$N_B$ = number of observations of branch formation from diVCA-Arp2/3-filament complexes.
$\tau_V^*$ = mean lifetime of diVCA on nascent branches.
$k_V^*$ = rate constant for diVCA release from the nascent branch.
$k_B$ = second order rate constant for branch formation (per mother filament subunit).

*2011*; *Xu et al., 2012*), and that alternative conformations of Arp2/3 complex (*Goley et al., 2004*; *Rodal et al., 2005*) may have different affinities for VCA.

From these experiments, we conclude that we have created a panel of mutants that modestly alter the association of diVCA with its binding partners. The V* and C* mutants, by disrupting interactions with Arp2/3 complex and monomeric actin, respectively, produce ternary diVCA-actin-Arp2/3 complexes that are less kinetically stable than wild-type. Conversely, the A* mutant produces a more stable ternary complex than wild-type.

## Rate of diVCA release from the nascent branch limits rate of daughter filament nucleation

The diVCA mutants have distinct activities in stimulating branch nucleation by Arp2/3 complex. In bulk solution (*Figure 5—figure supplement 1*) wild-type diVCA boosted Arp2/3 complex-dependent nucleation by 2.0 ± 0.5-fold (p=0.017; measured by concentration of filament barbed ends at the midpoint of the reaction, *Figure 5—figure supplement 1B*). The V* and C* mutants were more potent than wild-type (p=0.018 and 0.053, respectively), stimulating nucleation 10 ± 3-fold and 6 ± 3-fold. Conversely, the A* mutant was less potent than wild-type (p=0.040), stimulating only 1.4 ± 0.4 fold. In all of these experiments, the diVCA construct was at 25 nM, a concentration that produces near-maximal stimulation (*Figure 5—figure supplement 2A*).

Similar effects of the mutant diVCA constructs were seen in real-time observations of individual filaments being nucleated (*Figure 5A*). As expected, all three diVCAs were able to promote nucleation from the sides of existing filaments. More importantly, the mutant diVCA constructs shared with wild-type the key molecular behaviors discussed previously. Like wild-type, mutant diVCA constructs released from the nascent branch prior to initiation of the daughter filament (*Figure 5B*; *Figure 5—figure supplement 3*), and did not bind to Arp2/3 complex after the branch formed (*Figure 5C,D*; this later behavior could not be verified for the A* mutant because of its low activity in the experiments using tethered Arp2/3 complexes). Overall, these results suggest that the mutant constructs stimulate Arp2/3 complex by the same mechanism as wild-type. Moreover, the rank order of the diVCA construct nucleation activities was identical in the single-molecule measurements of the rate of branch formation observed on existing filaments (*Figure 5A*) and in bulk measurements of concentration of filaments generated in solution (*Figure 5—figure supplement 1B*). Thus, the mutants provide a range of activities both above and below wild-type that can be seen in both experimental modes, and the mechanism by which the mutants stimulate branch formation appears to be identical to wild-type.

Since high diVCA activity appears to correspond with weak binding to Arp2/3 complex, we next tested the hypotheses that the rate of diVCA release from the nascent branch quantitatively explains the nucleation activities of different mutants. While we already measured the dissociation rates of the diVCA constructs from Arp2/3 complex when the latter is not bound to the side of a mother filament (*Figure 4D*; *Table 1*), the rates of dissociation from the filament-bound Arp2/3 complex (the nascent branch) might be different.

Our observations (*Figure 2*) showed that once a nascent branch forms, it can have multiple fates (*Figure 6A*). Most often, the Arp2/3-diVCA complex simply dissociates from the filament (*Figure 6A*, thick red arrow). In a small fraction of the nascent branches, $f_{V-}$, diVCA departs first, leaving behind Arp2/3 complex bound to mother filament (*Figure 6A*, activated complex) where it may subsequently initiate a daughter filament. Furthermore, only a fraction of activated complexes subsequently formed a branch, so that the fraction of nascent branches that successfully produced a daughter filament, $f_B$, is less than $f_{V-}$. For some of the mutant constructs $f_{V-}$ or $f_B$ values (*Figure 6B*) could not be unambiguously distinguished from wild type in pairwise comparisons (p=0.05–0.15); others differed significantly (p=0.005–0.05) from wild-type (asterisks in *Figure 6B*).

We also measured the lifetimes of diVCA constructs on the nascent branches that ultimately produced daughter filaments. The mean lifetime ($\tau_V^*$, equal to the reciprocal of the sum of the rate constants for diVCA departure and Arp2/3 complex dissociation; 'Materials and methods') differed little between the different diVCA constructs (*Figure 6C*). Only the C* mutant had a significantly shorter mean lifetime on the nascent branch than wild-type diVCA (2 ± 1-fold, p=0.015), whereas the V* and A* mutants did not (p=0.17 and 0.48, respectively). This observation suggests ('Materials and methods') that the mutations do not substantially alter the rate of the main pathway of nascent complex breakdown, which is the dissociation of the intact Arp2/3-diVCA complex from filament sides

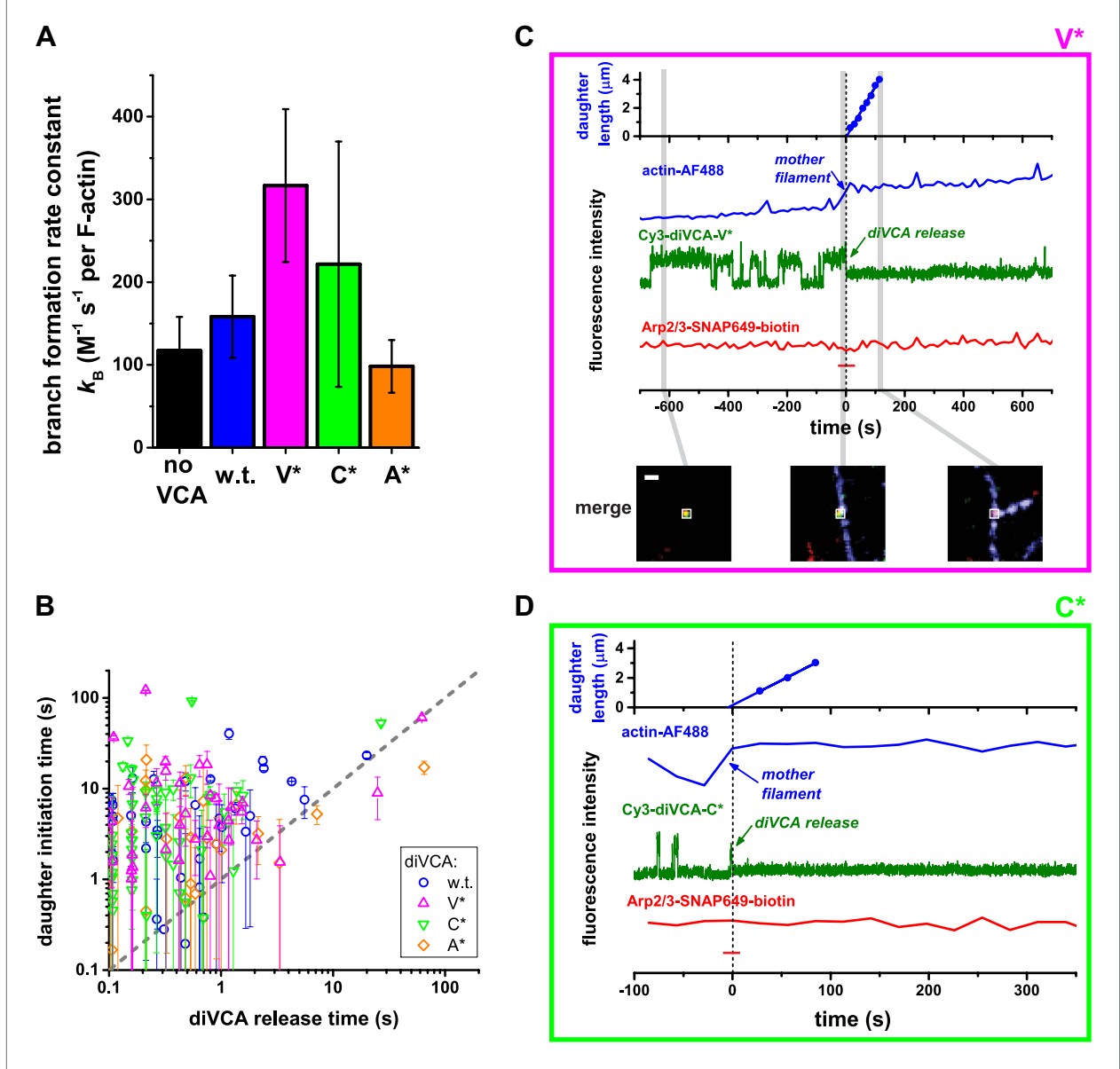

**Figure 5**. diVCA constructs differ in the rate but not the pathway of activity in stimulating branch formation. (**A**) Rate (±S.E.) of initiation of daughter filament growth by Arp2/3 complex in the absence or presence of diVCA wild-type and mutant constructs. $k_B$, the second order rate constant for the appearance of branches on existing filaments, per subunit, was calculated from observations of branch formation on existing filaments, as in **Figure 2** ('Materials and methods'). (**B**) Comparison of the time (±S.E.) of daughter filament initiation with the time of diVCA release from the nascent branch for wild-type (data replotted from **Figure 2E**) and mutant constructs (**Figure 5—figure supplement 3**). (**C** and **D**) Example records showing the length of a nucleated daughter filament and the fluorescence intensity from actin, diVCA, and individual tethered Arp2/3 complex molecules, as in **Figure 3**. Mutant Cy3-diVCA was 0.5 nM V* in C, or 1.0 nM C* in D. The merged fluorescence images in **C** were recorded at the indicated times and the white squares mark the area from which the fluorescence was integrated to produce the intensity records. Scale bar, 1 µm. Both mutants bound readily to tethered Arp2/3 prior to but not after branch formation.

The following figure supplements are available for figure 5:

**Figure supplement 1**. Analysis of diVCA mutant activities in bulk actin polymerization assays.

**Figure supplement 2**. Saturation of stimulation of Arp2/3 complex actin nucleation activity by diVCA constructs.

**Figure supplement 3**. Relationship relation between diVCA release from the nascent branch and daughter filament initiation for each of the three mutant diVCA constructs.

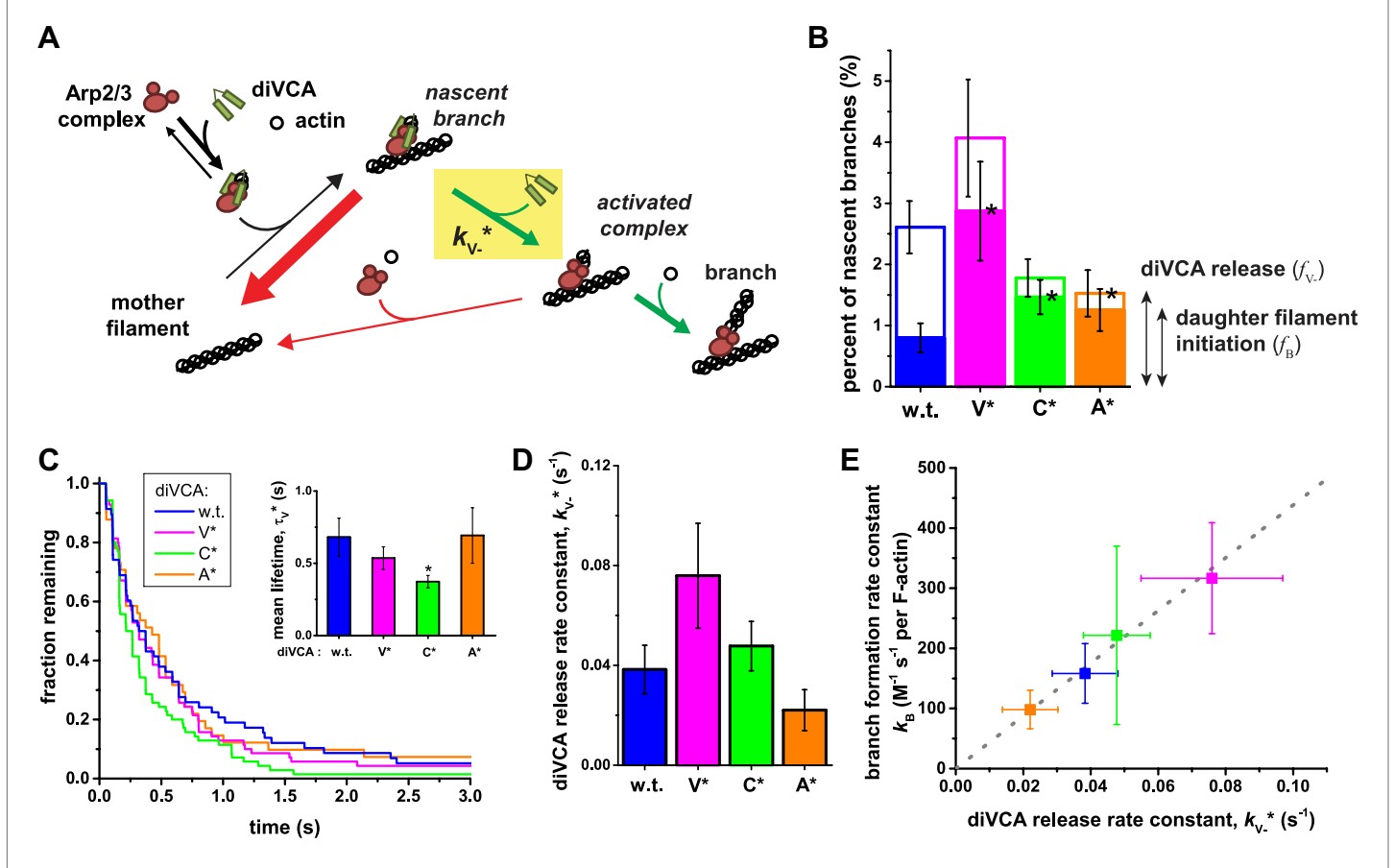

**Figure 6**. Release of diVCA from nascent branches is rare and limits the rate of daughter nucleation. (**A**) Schematic mechanism of diVCA stimulated branch formation (see text). The key activation step, release of diVCA from the nascent branch, is highlighted. (**B**) Classification of nascent branch fates observed in single molecule experiments (e.g., **Figure 2B and C**; **Figure 2—figure supplement 3**). Overall bar height indicates the fraction (±S.E.) of nascent branches that release diVCA leaving behind a filament-bound Arp2/3 complex. Filled bar height shows the fraction (±S.E.) of nascent branches that nucleate a daughter filament. (**C**) Cumulative lifetime distributions of diVCA molecules on the subset of filament-bound Arp2/3 complexes observed to produce branches in single-molecule experiments. Inset: mean lifetimes (±S.E.). (**D**) Rate constants (±S.E.) for diVCA dissociation from the nascent branch, calculated from the mean lifetimes in **C** and release efficiencies in **B**. (**E**) Correlation between the rate constant of diVCA-stimulated Arp2/3 complex branch nucleation (from **Figure 5A**) and the rate constant of diVCA release from the nascent branch (from **D**). Correlation coefficient $r = 0.9928$ is unlikely to arise by coincidence (p=0.0045). Dotted line is a linear fit constrained to pass through the origin.

The following figure supplements are available for figure 6:

**Figure supplement 1**. Correlation between the actin nucleation activity of Arp2/3 activated by wild-type and mutant diVCA constructs (from **Figure 5—figure supplement 1B**) and the rate constant of diVCA release from the nascent branch (from **Figure 4F**).

(**Figure 6A**, thick red arrow). Instead, the mutations principally affect the fraction of nascent complexes that release diVCA, and are productive in forming branches.

Based on the measurements of **Figure 6B,C**, we can calculate the rate constant for dissociation of wild-type and mutant diVCAs from the nascent branch (**Figure 6A**, $k_{V-}^*$) as $f_{V-}/\tau_V^*$ (**Figure 6D**). Strikingly, this rate of release of diVCA from the nascent branch is proportional, within experimental uncertainty (**Figure 6E**), to the branch formation rate constant $k_B$ measured above (**Figure 5A**; **Figure 6—figure supplement 1**). Despite the relatively subtle effects of the mutants on the measured rate constants, the high correlation coefficient ($r = 0.993$) is unlikely to be coincidental (p=0.0045; 'Materials and methods'). This relationship provides strong support for the hypotheses that diVCA release is a prerequisite for daughter filament formation, and that the rate of release of diVCA limits the rate at which nascent branches initiate daughter filament growth.

## Discussion

By placing three different colors of fluorescent labels on Arp2/3 complex, diVCA, and actin, we directly observed the sequence and kinetics of key steps in the branch nucleation pathway. The observations confirm our previous results (*Smith et al., 2013*) that branch nucleation is inefficient even at near-saturating VCA protein concentrations, with only a small fraction of Arp2/3 complex-mother filament associations yielding branches. In contrast to the tight binding of diVCA to Arp2/3 complex in solution, we show that diVCA binding is undetectable when Arp2/3 complex is incorporated into a branch junction.

Significantly, we observed that the branch formation process is strictly dependent on release of diVCA from the filament-bound Arp2/3-diVCA complex. Taken together with earlier work, the data support a specific mechanism for diVCA-stimulated actin nucleation by Arp2/3 complex in which diVCA plays a dual role (*Figure 6A*). Initially, diVCA stimulates assembly of the nascent branch by associating tightly with Arp2/3 complex and actin monomers, which promotes the binding of Arp2/3 complex to the sides of filaments. However, once the nascent branch forms, diVCA plays an inhibitory role: diVCA must dissociate before daughter filament can grow. In mutants that alter the interactions of diVCA with Arp2/3 complex or monomeric actin, alterations of the rate at which diVCA leaves the nascent branch exactly parallel changes in the efficiency of branch formation. These observations strongly suggest that diVCA dissociation is the key, rate-limiting step in daughter filament nucleation.

The key mechanistic conclusions reached here with dimerized VCA constructs in vitro are likely to apply to the mechanism of activation of branch nucleation by native WASP oligomers in vivo. Data on dimeric VCA showing a minimum crosslinker length for high activity (*Padrick et al., 2011*) strongly suggests that the system passes through a form with two VCAs bound at some point during nucleation. The crosslinkers and spacer sequences used here allow a maximum VCA separation of 122 Å (*Padrick et al., 2011*). During Arp2/3 complex activation in diverse contexts in living cells, WASP family proteins are separated by similar or even smaller distances. For example, when dimerized by activators such as EspFu, the WASP protein VCA regions would be ~80 Å apart if the proline rich region is considered as a random coil polymer (*Padrick et al., 2008*). Similarly, ActA density on the surface of *Listeria* separates its VCA-like sequences by 19 nm, close enough to function as VCA dimers (*Footer et al., 2008*). N-WASP proteins recruited to moving PIP2-rich vesicles have a high local density, such that their average spacing is <50 Å (*Co et al., 2007*), and in rocketing vesicles, N-WASP proteins are recruited by Nck onto the vesicle surface such that their average separation is ~100 Å (*Ditlev et al., 2012*). Bzz1 and Cdc15 are the activators of the fission yeast WASP family protein Wsp1 during endocytosis, and they array their SH3 domains (which engage Wsp1) at a density sufficient to bring VCA regions from adjacent Wsp1 proteins within 50–80 Å (*Arasada and Pollard, 2011*).

The reaction scheme in *Figure 6A* describes the essential features of the Arp2/3 complex-dependent filament nucleation process. This mechanism, derived from direct observation of single molecules as opposed to fitting of bulk data, is broadly consistent with proposed kinetic schemes for Arp2/3 complex nucleation of filaments (*Zalevsky et al., 2001*; *Beltzner and Pollard, 2008*; *Smith et al., 2013*). The dominant pathway is that VCA and actin monomer associate with Arp2/3 complex in solution, and this complex then binds to an existing filament, after which an activation step occurs, which allows the daughter filament to elongate. Here we add two informative points. First, we ascribe a distinct mechanism to the activation step, the release of diVCA. Previously this step had been ascribed to structural rearrangements within the Arp2/3 complex (*Dayel et al., 2001*; *Zalevsky et al., 2001*; *Beltzner and Pollard, 2008*). Second, our measurements indicate that most engagements of diVCA-Arp2/3 complexes with the mother filament are resolved by dissociation of the complex from filament without releasing VCA, an idea distinct from previous models (*Beltzner and Pollard, 2008*). It is possible that the observed inefficiency in branch formation is to allow for positive regulation of branch formation by factors not present in our experiment (*Smith et al., 2013*). Consistent with this idea, cortactin has been recently demonstrated to accelerate release of WASP proteins from Arp2/3 complex (*Helgeson and Nolen, 2013*). Since WASP proteins are tethered to the cell membrane, it is also possible that mechanical tension between the filament network and the membrane plays a similar role in promoting WASP release and consequent daughter growth, which might allow alteration of cell motility in response to mechanical stimuli.

While our scheme encompasses key features of the mechanism of diVCA stimulation of Arp2/3 complex-mediated branch formation, it should be noted that the scheme shown in *Figure 6A* is

not complete. The clearest indication of this is that the diVCA lifetime distribution observed in *Figure 2D* is multi-exponential, whereas the scheme of *Figure 6A* predicts only a simple single-exponential distribution. The data can be explained if there are two or more conformations of filament-bound diVCA-Arp2/3 complexes that differ in kinetic stability. This proposal is consistent with our previous kinetic analysis with monomeric VCA (*Smith et al., 2013*) and with demonstrations that Arp2/3 complex exists in multiple conformations in solution (*Goley et al., 2004*; *Rodal et al., 2005*; *Xu et al., 2012*). Formulating a more complete kinetic mechanism of diVCA stimulation that accounts for the multiple diVCA-Arp2/3 complex conformations will require additional data.

Our model in which association of VCA with the nascent branch inhibits initiation of daughter filament growth is consistent with the known interaction of the WH2 motif (V-region) with actin in a conserved cleft involved in longitudinal filament contacts (*Hertzog et al., 2004*; *Irobi et al., 2004*; *Chereau et al., 2005*). We suspect that inhibition of daughter filament elongation from nascent branches results from the WH2 motif staying engaged with the cleft, occluding addition of the next actin subunit in the daughter filament (*Egile et al., 1999*; *Higgs et al., 1999*). Further support for this model is realized by recent reports showing that VCA peptides covalently crosslinked to actin monomers are inactive in stimulating Arp2/3 complex-mediated actin nucleation (*Boczkowska et al., 2008*; *Ti et al., 2011*). Moreover, the C-region of VCA likely occupies a similar location on Arp2 and Arp3 that the V-region occupies on actin, preventing insertion of subdomain 2 of the actin recruited by the V-region into the cleft on the bound Arp subunit (*Hertzog et al., 2004*; *Irobi et al., 2004*; *Chereau et al., 2005*; *Boczkowska et al., 2008*; *Padrick et al., 2011*; *Ti et al., 2011*). Thus, the molecular structures are consistent with the observed dual function of VCA in both stimulating daughter nucleation (by recruiting actin monomers to Arp2/3 complex) and in suppressing daughter nucleation (by blocking assembly of the daughter filament). These structural models are also consistent with the low affinity of diVCA for branch junctions. If neither the V-region nor the C-region can bind the branch (when the actin and Arp clefts are occupied by D-loops of daughter filament actins), then the affinity of VCA for the branch may be of the order of that observed for isolated A-region. This affinity has been reported to be approximately 9 µM (*Marchand et al., 2001*), consistent with the lack of observed binding under the conditions necessary for single molecule observations.

In cells, active WASP proteins are predominantly tethered to membranes. Taken together, our data suggests a straightforward mechanism by which WASP regulation of Arp2/3 complex can cause branches to form preferentially at membranes without having membrane attachment restrict network growth (*Figure 7*). Binding of Arp2/3 complex to VCA dimers in the absence of mother filaments is of high affinity and of long lifetime, allowing dimerization of WASP by upstream activators to promote recruitment of Arp2/3 complex to the membrane surface (*Figure 7*, 'WASP dimer association with Arp2/3 complex'). Association with VCA promotes filament binding by Arp2/3 complex ('Arp2/3 complex engagement of filament'), but that process is readily reversible and the nascent branch most frequently is simply lost through dissociation ('Filament release'). More rarely, WASP detaches ('WASP release') leaving Arp2/3 complex associated with the mother filament. This severs the direct linkage to the membrane, and only then allows the daughter filament to nucleate and grow ('Initiation of daughter filament elongation'). Our data show that once the branch forms, WASP does not rebind. The dissociation of WASP prior to nucleation and the lack of rebinding provide an appealing explanation for how WASP dimers stimulate branched network formation at the membrane without interfering with network growth.

## Materials and methods

### Protein design, labeling, and purification

The bis-maleimide crosslinked N-WASP VCA dimers were prepared from human N-WASP VCA (amino acid residues 432–505), with the sequence CGGSGGSGGSGGS appended at the N-terminus. This sequence was generated by PCR using overlapping primers encoding the N-terminal extension. The resulting PCR product was cloned into a plasmid derived from pGEX2T with a TEV protease site between the sequence for GST and the multiple cloning site (which was modified to include a 5′ NdeI site and 3′ BamHI site). Mutations were introduced by designing overlapping primers containing the desired mutations, which were used in PCR amplification of the region 5′ to the modification and 3′ to the modification. These products were used as template in a second round of PCR that fused them into a single cassette containing the desired mutation. The resulting product was digested with NdeI and BamHI and ligated into the above described expression vector. Cloned products were verified by DNA sequencing.

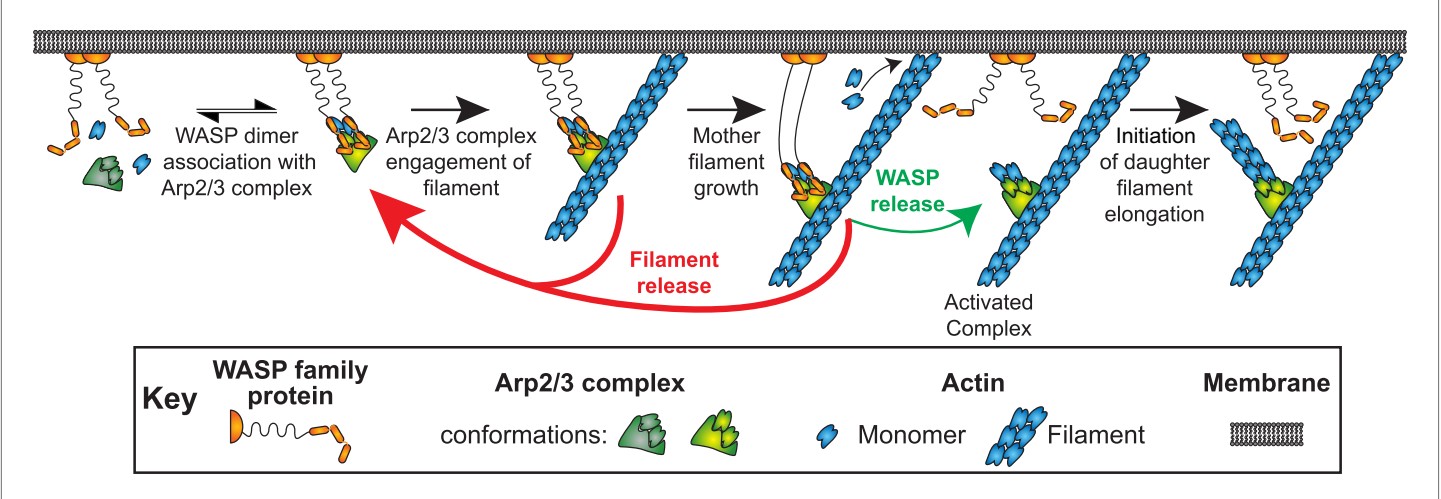

**Figure 7**. Model of WASP-Arp2/3 complex stimulated actin branch formation at cell membranes (see text).

Crosslinked VCA dimers were prepared using a common strategy. Proteins were expressed in *Escherichia coli* BL21(DE3) T1[R] cells using IPTG induction at 37°C for 3 hr. Cells were harvested, resuspended in 25 ml of buffer (20 mM Tris pH 8.0, 50 mM NaCl, 2 mM EDTA, 1 mM DTT and 1 mM PMSF) per l of culture, and frozen at −80°C until needed. Cells were lysed using a cell disruptor and clarified by centrifugation at 19,500 rpm in a JA25.50 rotor (Beckman Coulter Inc., Brea, CA). Clarified lysate was purified using DEAE Sepharose FF (GE Healthcare Biosciences, Pittsburgh, PA) ion exchange chromatography, followed by Glutathione Sepharose (GE Healthcare Biosciences) chromatography, followed by cleavage with TEV protease. Cleaved VCA peptides were purified by SOURCE 15Q ion exchange chromatography. Next, the pooled VCA was concentrated and the DTT removed, by dilution and passing over a 0.5 ml SOURCE 15Q column in buffer lacking reducing agent. BMCy3 (#C959070, Toronto Research Chemicals, Toronto, Canada) was prepared as a 20 mM stock in anhydrous DMSO. VCA materials were quantified by absorption at 280 nm, and BMCy3 was added to a final concentration of one equivalent (about 80 μM depending on the preparation), split over three additions separated by 10 min incubations at room temperature. After a final 20 min incubation at room temperature, the reaction was quenched with 2 mM DTT, and the reaction was purified using SOURCE 15 Q ion exchange chromatography, followed by Superdex 200 gel filtration chromatography (GE Healthcare Biosciences). Purification was tracked using SDS-PAGE analysis, and concentrations were determined from absorption at 552 nm, using an extinction coefficient of 150,000 M$^{-1}$ cm$^{-1}$.

GST-VCA and GST-VVCA were produced from vectors lacking the CGGSGGSGGSGGS N-terminal extension. Expression and purification followed a similar protocol as for the crosslinked VCA dimers, with the modification that GST was not cleaved from the product following Glutathione Sepharose purification. Purification continued with SOURCE 15Q ion exchange chromatography and Superdex 200 gel filtration chromatography. Concentrations were measured using absorbance at 280 nm.

The fluorescence anisotropy probe for actin binding, VCA-AF488, was produced from a similar vector encoding N-WASP VCA amino acid residues 430–505 (with the introduced mutations S430C and C431A, and lacking the CGGSGGSGGSGGS extension). Expression and purification was the same as for the VCA dimers, through the SOURCE 15Q step in buffer without reducing agent. At this point, 1 ml of 40 μM VCA was labeled with 150 μM AlexaFluor488 maleimide (#A-10254, Molecular Probes/ Life Technologies, Grand Island, NY, 40 mM stock in anhydrous DMSO). After 2 hr reaction at room temperature, the reaction was quenched with 2 mM DTT and VCA-AF488 was purified by SOURCE15Q and Superdex 75 chromatography. Labeling efficiency was judged to be nearly 100% from a shift in mobility by SDS-PAGE (small shift judged using a reference of flanking unlabeled material). Concentration was measured as the absorbance at 492 nm, using an extinction coefficient of 71,000 M$^{-1}$ cm$^{-1}$.

The fluorescence anisotropy probe for Arp2/3 complex binding, VVCA-A462C-A594, was produced by using a similar method to VCA-AF488 with two modifications. First, the vector used encoded N-WASP VCA amino acid residues 393–505, with two mutations, C431A and A462C. Second, the VCA

was labeled with AlexaFluor 594 maleimide (#A-10256, Molecular Probes/Life Technologies, 40 mM stock in anhydrous DMSO), instead of AlexFluor 488 maleimide. Quantification was performed using absorbance at 588 nm and an extinction coefficient of 96,000 $M^{-1}$ $cm^{-1}$. The fluorescence anisotropy probe used for examining simultaneous binding of yeast Arp2/3 complex and *Drosophila* 5C actin, VCA-AF594, was produced by using a similar method to VCA-AF488, but substituting the AlexaFluor 594 maleimide (mentioned above) for AlexaFluor 488 maleimide.

*S. cerevisiae* Arp2/3 complex (used in *Figure 2—figure supplements 2 and 4*, *Figure 5—figure supplement 2*) was purified from commercial baker's yeast (#05020, Red Star Yeast Company, Milwaukee, WI) using a method adapted from published protocols (*Egile et al., 1999*; *Lechler et al., 2001*) with added SOURCE 15Q and Superdex 200 chromatography steps (*Doolittle et al., 2013b*). Endogenous bovine Arp2/3 complex (used in *Figure 2—figure supplement 1*, *Figure 4—figure supplement 1B*) was purified from calf thymus using previously described methods (*Higgs et al., 1999*; *Doolittle et al., 2013c*). SNAP-tagged Arp2/3 complex was purified from recombinant *S. cerevisiae* and labeled as previously described (*Smith et al., 2013*), except that labeling used BG-649 and BG-649-PEG-biotin ('Synthesis of SNAP-tag substrates BG-649 and BG-649-PEG-biotin' below) to yield Arp2/3-SNAP649 and Arp2/3-SNAP649-biotin.

Rabbit muscle actin, pyrene-labeled actin, AlexaFluor488-labeled actin, and biotinylated actin were purified as described (*Spudich and Watt, 1971*; *Smith et al., 2013*). Non-polymerizable *Drosophila melanogaster* 5C actin was prepared according to established methods (*Joel et al., 2004*), but with the mutation D287A/V288A/D289A instead of A204E/P243K. Characterization of this mutation will be described elsewhere (*Zahm et al., 2013*).

## Synthesis of SNAP-tag substrates BG-649 and BG-649-PEG-biotin

In the syntheses, commercially available compounds were used without further purification and reaction yields are not optimized. Reversed-phase high-performance liquid chromatography (HPLC) was performed on Agilent LC/MS Single Quad System 1200 Series (analytical) and Agilent 1100 Preparative-scale Purification System (semi-preparative). Analytical HPLC was performed on Waters Atlantis T3 C18 column (2.1 × 150 mm, 5 µm particle size) at a flow rate of 0.5 ml/min with a binary gradient from Phase A (0.1 M triethyl ammonium bicarbonate [TEAB] or 0.1% trifluoroacetic acid [TFA] in water) to Phase B (acetonitrile) and monitored by absorbance at 280 nm. Semi-preparative HPLC was performed on VYDAC 218 TP series C18 polymeric reversed-phase column (22 × 250 mm, 10 µm particle size) at a flow rate of 20 ml/min. Mass spectra were recorded by electrospray ionization (ESI) on an Agilent 6210 Time-of-Flight (TOF) or 6120 Quadrupole LC/MS system.

BG-649 was prepared by reacting the building block BG-NH$_2$ (New England Biolabs, Ipswich, MA) with the dye *N*-hydroxysuccinimide ester DY-649 NHS (Dyomics GmbH, Jena, Germany) as described previously (*Keppler et al., 2004*). BG-NH$_2$ (0.54 mg, 2.0 µmol) was dissolved in anhydrous DMF (0.5 ml). DY-649 NHS (2.0 mg, 2.0 µmol) and triethylamine (0.4 µl, 3.0 µmol) were added and the reaction mixture stirred overnight at room temperature. The solvent was removed under vacuum and the product purified by reversed-phase HPLC using 0.1 M TEAB/acetonitrile gradient. Yield: 74%. BG-649: ESI-MS *m/z* 1095.2 [M-H]⁻ (calculated for $C_{48}H_{56}N_8O_{14}S_4$, *m/z* 1095.3).

BG-649

The bifunctional BG-649-PEG-Biotin (which includes both a DY-649 dye and a biotin moiety) was prepared by successive couplings of commercially available α-*N*-Fmoc-ε-*N*-Dde-lysine (Merck KGaA, Darmstadt, Germany) with BG-NH$_2$ (New England Biolabs, Ipswich, MA), *N*-(+)-biotin-6-aminocaproic acid *N*-succinimidyl ester (Sigma-Aldrich, St. Louis, MO) and DY-649 NHS (Dyomics GmbH, Jena,

Germany) according to synthetic route described previously (*Kindermann et al., 2004*). BG-649-PEG-Biotin was synthesized as follows: BG-NH₂ (250.0 mg, 0.92 mmol) was dissolved in anhydrous DMF (8 ml). HBTU (*N,N,N',N'*-Tetramethyl-O-(1H-benzotriazol-1-yl)uronium hexafluorophosphate) (368.0 mg, 0.97 mmol), triethylamine (135 μl, 0.97 mmol), and Fmoc-Lys(Dde)-OH (515.5 mg, 0.97 mmol) were added and the reaction mixture stirred overnight at room temperature. The reaction mixture was poured onto water (80 ml). The white solid was collected by filtration, washed twice with water, and dried in dessicator under vacuum overnight. Yield: 91%. BG-Lys(Dde)-Fmoc (50 mg, 63.7 μmol) was dissolved in anhydrous in DMF (5 ml). Et₂NH (19.8 μl, 191.1 μmol) was added and the reaction mixture stirred overnight at room temperature. The solvent was removed under vacuum for 6 hr and the residue dissolved in DMF (3 ml). Fmoc-12-amino-4,7,10-trioxadodecanoic acid (29.7 mg, 66.9 μmol), triethylamine (26.6 μl, 191.1 μmol) and HBTU (36.3 mg, 95.6 μmol) were added and the reaction mixture stirred for 1 hr at room temperature. The reaction completion was monitored by LC/MS. The solvent was removed under vacuum and the product purified by reversed-phase HPLC using 0.1 M TEAB/acetonitrile gradient. Yield: 50%. BG-Lys(Dde)-PEG-NHFmoc: ESI-MS *m/z* 988.4 [M+H]⁺ (calculated for $C_{53}H_{65}N_9O_{10}$, *m/z* 988.5). BG-Lys(Dde)-PEG-NHFmoc (31.6 mg, 31.9 μmol) was dissolved in anhydrous DMF (2 ml). Et₂NH (9.9 μl, 95.7 μmol) was added and the reaction mixture stirred overnight at room temperature. The solvent was removed under vacuum for 6 hr and the residue dissolved in DMF (2 ml). *N*-(+)-biotin-6-aminocaproic acid NHS (14.5 mg, 31.9 μmol) and triethylamine (13.3 μl, 95.7 mmol) were added and the reaction mixture stirred overnight at room temperature. The reaction completion was monitored by LC/MS. A 2% solution of hydrazine in DMF (0.5 ml) was added and the reaction mixture stirred for 1 hr at room temperature. The solvent was removed under vacuum and the product purified by reversed-phase HPLC using 0.1% TFA in water/acetonitrile gradient. Yield: 75%. BG-Lys(NH₂)-PEG-Biotin: ESI-TOFMS *m/z* 939.4873 [M-H]⁻ (calculated for $C_{44}H_{68}N_{12}O_9S$, *m/z* 939.4880). BG-Lys(NH₂)-PEG-Biotin (2.3 mg, 2.13 μmol) was dissolved in anhydrous DMF (1 ml). DY-649 NHS (2.1 mg, 2.13 μmol) and triethylamine (0.45 μl, 3.2 μmol) were added and the reaction mixture stirred overnight at room temperature. The solvent was removed under vacuum and the product purified by reversed-phase HPLC using 0.1 M TEAB/acetonitrile gradient. Yield: 73%. BG-649-PEG-Biotin: ESI-TOFMS *m/z* 882.3166 [M-2H]²⁻ (calculated for $C_{79}H_{110}N_{14}O_{22}S_5$, *m/z* 882.3188).

BG-649-PEG-Biotin

## Actin assembly kinetics

Actin assembly kinetics measurements were performed in a fashion similar to previously published (*Cooper et al., 1983*; *D'Agostino and Goode, 2005*; *Padrick et al., 2008*, *2011*; *Doolittle et al., 2013a*). Briefly, a rabbit muscle actin stock (5% pyrene labeled in *Figure 5—figure supplement 1* or 10% pyrene labeled in *Figure 2—figure supplement 1 and 2*, and *Figure 5—figure supplement 2*) in buffer G (2 mM Tris pH 8.0, 200 μM CaCl₂, 1 mM NaN₃, 100 μM ATP, 0.5 mM DTT) was combined with 1/10th volume 10 mM EGTA, 1 mM MgCl₂, and then with enough buffer G-Mg (same as buffer G but substituting MgCl₂ for CaCl₂) to dilute the overall actin concentration to 4 μM. After a 2 min incubation in this buffer, the actin solution was combined with an equal volume of Arp2/3 complex and VCA materials in double strength KMEI buffer (such that the final solution was 10 mM imidazole pH 7.0,

50 mM KCl, 1 mM EGTA, 1 mM MgCl$_2$, 0.5 mM DTT, with one half concentration of buffer G carrying over from the actin solution). For experiments designed to enable quantitative comparison of actin assembly rates (*Figure 5—figure supplement 1*) to results from single molecule analysis (*Figure 6D*, *Figure 6—figure supplement 1*) the buffer was supplemented with additional components (final concentrations: 10 mM DTT, 0.2 mM ATP, 15 mM glucose, 0.02 mg/ml catalase, 0.1 mg/ml glucose oxidase, 0.1% bovine serum albumin, 1 mM 6-hydroxy-2,5,7,8-tetramethylchroman-2-carboxylic acid (Trolox), 1 mM 4-nitrobenzyl alcohol, and 0.5 mM propyl gallate) so as to maintain similar buffer conditions in the two experiments. Reactions were then immediately placed in a cuvette in a PTI Quantamaster spectrofluorometer. Pyrene-actin fluorescence was observed over time by exciting at 365 nm and observing at 407 nm. For some assays (*Figure 2—figure supplement 1 and 2*, *Figure 5—figure supplement 2*), 10% pyrene labeling was used, and the reactions were placed into 96-well plates and followed using a plate reader (VarioSkan Flash, Thermo Scientific, Hudson, NH).

Actin filament barbed end concentrations were evaluated at 50% polymerization ($t_{50}$) by first scaling the pyrene fluorescence intensity over the full range of filament concentrations (0–1.9 µM), then fitting the slope (actin assembly rate) between 42% and 58% polymerization. These rates were then divided by the approximate filament elongation rate (10 subunits per second) to obtain the barbed end concentrations (*Figure 5—figure supplement 1B*). Fold stimulation of branch formation by diVCA was calculated by subtracting the barbed ends created by actin alone from the total barbed ends formed in the presence of Arp2/3 complex, then dividing the concentrations of these excess barbed ends formed in the presence diVCA by those formed in the absence of diVCA.

## Fluorescence anisotropy

Binding of VCA to actin was monitored by fluorescence anisotropy (*Figure 2—figure supplement 4*, *Figure 4B*). 20 nM N-WASP VCA-AF488 was mixed with the indicated concentrations of actin and Cy3-diVCA competitor with buffer additions to bring the final mixture to 10 mM imidazole pH 7.0, 50 mM KCl, 1 mM EGTA, 1 mM MgCl$_2$, 0.5 mM DTT, 0.1 mM ATP, and 1/10th residual concentration of buffer G. The mixtures were incubated for 3 min at room temperature prior to placing in a 3 mm by 3 mm cuvette in a T-form PTI Quantamaster Spectrafluorometer, equipped with Glan-Thompson polarizers. Emission intensity was averaged for 3 min and converted to anisotropy values, after correcting for background signal intensity and G-factor. Competition binding experiments (including N-WASP diVCA competition) were performed with 20 nM N-WASP VCA-AF488, 200 nM rabbit muscle actin, and the indicated concentrations of VCA dimers. Binding isotherms (both direct and competition binding) were fit to a complete competition binding solution for a single site receptor, using Levenberg-Marquardt nonlinear least squares methods, with bound and free fluorescence anisotropy as fit parameters. Direct binding isotherms were fit with the concentration of competitor ligand set to zero. Fitting of the competition-binding isotherm used the direct binding $K_D$ that was obtained from fitting the direct binding isotherm. Fit values for free fluorescence anisotropy values were similar to the actual free anisotropy, and the values of free and bound fluorescence anisotropy determined from competition binding experiments was similar to that of the direct binding system.

Binding of VCA to bovine Arp2/3 complex in solution (*Figure 2—figure supplement 1D–E*, *Figure 4—figure supplement 1B*) was monitored by fluorescence anisotropy using the same basic protocol as for actin binding, with a few modifications. First, the reporter was 20 nM VVCA-A462C-AF594. Next, as there was no actin present, there was no residual buffer G in the mixture. Finally, the mixture was incubated for 10 min prior to measurement, and the fluorescence intensity data averaged for 5 min. Solution binding of VCA to non-polymerizable *Drosophila* 5C actin, and to yeast Arp2/3 complex, was followed by fluorescence anisotropy. Acquisition and processing was similar to the actin binding assay described above, except 10 nM VCA-A594 was used, and either *Drosophila* 5C actin was added or endogenous budding yeast Arp2/3 complex was added. From fitting the Arp2/3 complex titration curve, 300 nM was judged to have essentially complete binding, and this was added to a separate titration of actin with Arp2/3 complex present. In fitting all three data sets, a single binding site was assumed on VCA.

## Colocalization single molecule spectroscopy (CoSMoS)

Single molecule imaging was performed on a custom built multi-wavelength total internal reflection fluorescence (TIRF) microscope, described previously (*Friedman et al., 2006*; *Hoskins et al., 2011*; *Friedman and Gelles, 2012*; *Smith et al., 2013*). Briefly, the microscope design permitted selective

fluorescence excitation of molecules immobilized on the surface of a glass flow chamber (*Smith et al., 2013*) using three lasers at wavelengths 488 nm, 532 nm, and 633 nm. Emissions were split into short (<635 nm) and long (>635 nm) wavelengths and focused on different locations on the camera, to allow for simultaneous acquisition of two-color fluorescence (*Friedman et al., 2006*). Experiments were performed using two different modes of operation described below, one for recording Arp2/3 complex and diVCA interacting with tethered filaments, and another for recording diVCA and filaments interacting with tethered Arp2/3 complexes.

All experiments were performed in TIRF buffer: 50 mM KCl, 1 mM $MgCl_2$, 1 mM EGTA, 10 mM imidazole pH 7.2, 10 mM DTT, 0.2 mM ATP, 15 mM glucose, 0.02 mg/ml catalase, 0.1 mg/ml glucose oxidase, 0.1% bovine serum albumin (BSA), and 2% dextran. To suppress blinking of the fluorophores, TIRF buffer was supplemented with a mixture of triplet state quenchers: 1 mM 6-hydroxy-2,5,7,8-tetramethylchroman-2-carboxylic acid (Trolox), 1 mM 4-nitrobenzyl alcohol, and 0.5 mM propyl gallate, which were dissolved in DMSO at a 200× stock concentration. Prior to flow chamber assembly (*Smith et al., 2013*), surfaces were first cleaned by sonication in detergent (2% Micro-90, 1 hr), ethanol (1 hr), KOH (0.1 M, 30 min), and deionized water (10 min), then coated with a mixture of methoxy-poly (ethylene glycol)-silane and biotin-poly (ethylene glycol)-silane (mPEG-sil-2000 and biotin-PEG-sil-3400; Laysan Bio Inc., Arab, AL) in 80% ethanol, pH 2 HCl, baked overnight at 70°C. Immediately before each experiment, the chambers were incubated in 0.5 mg/ml BSA and 0.03 mg/ml streptavidin in successive washes.

The oxygen scavenging activity of the glucose/glucose oxidase/catalase system in our buffers resulted in a gradual decrease in pH over time through the course of our experiments (*Shi et al., 2010*). We measured this pH change in our microscope flow chambers using a pH-sensitive fluorophore (SNARF-4F, pKa ~6.4; Molecular Probes/Life Technologies), by recording the ratio of fluorescence emission intensities in the two color channels of our microscope, exciting at fixed wavelength (533 nm). The ratiometric response from 0.1 μM SNARF-4F was first calibrated using strong buffers (0.1 M phosphate buffer) over a pH range of 5.8–8.0 (increment 0.2). The time dependent pH of our TIRF buffer was then measured and found to decrease from 7.0 to 6.5 over the typical time course of our experiments (~30 min). The effect of pH on Arp2/3 complex mediated actin filament assembly was also tested and showed that decreasing the pH from 7 to 6 increased nucleation twofold without VCA, while stimulation by diVCA and mutants differed by <1.5-fold. Thus, we expect that the buffer conditions used for the single molecule experiments had only a minor influence on diVCA stimulation of branch formation by Arp2/3 complex.

For tethered-filament experiments, a mixture of 10% AF488 labeled actin, 1% biotinated actin, and unlabeled actin was allowed to polymerize for 2–8 hr at 3 μM in TIRF buffer without dextran. Preassembled filaments were then diluted 40-fold, flowed into the microscope observation chamber coated with a 1:100 mixture of biotin-PEG-silane:PEG-silane, and allowed to adhere to the surface. The chamber was then rinsed with TIRF buffer and the reaction mixture was introduced: 1 μM actin (10% AF488 labeled), 5 nM Arp2/3-SNAP649, and 5 nM Cy3-diVCA. Data was recorded for 15–30 min in cycles of a single 50 ms frame of fluorescence emission using 488 nm laser excitation (to image actin filaments) followed by continuous acquisition of 200 frames (50 ms per frame) of emissions using dual 532 nm and 633 nm excitation (to image Arp2/3 complex and diVCA). Each cycle was followed by a ~1 s delay during which the focus was automatically adjusted (*Smith et al., 2013*), bringing the total cycle interval to ~12 s.

For tethered-Arp2/3 experiments, 1 nM Arp2/3-SNAP-649-biotin was introduced into a microscope observation chamber coated with a 1:2000 mixture of biotin-PEG-silane:PEG-silane, and allowed to adhere to the surface. The chamber was then rinsed with TIRF buffer and the reaction mixture was introduced: 1 μM actin (10% AF488 labeled) and 0.1–1 nM Cy3-diVCA. Data was recorded for 30–50 min in cycles of a single frame of fluorescence emission using dual 488 nm and 633 nm excitation (to image actin filaments and tethered-Arp2/3) followed by continuous acquisition of emissions using 532 nm excitation (to image VCA). For these experiments the frame duration was varied in the range 0.05–1 s and the number of frames per cycle was adjusted such that each cycle interval was 14–28 s. The frame duration was increased to allow for the use of lower power excitation, so as to characterize and correct for the effect of photobleaching on the observed lifetimes of Cy3-diVCA bound to tethered Arp2/3 complexes (*Figure 4—figure supplement 2D*).

For each experiment, we calibrated the transformation matrix for superimposing images from the split emission recordings by imaging fluorescent beads that emitted in both the short- and

long-wavelength color channels of the microscope. This transformation corrected for the translocation of the two color images on the camera, as well as differences in the rotation and magnification in the two channels. For records of colocalization on tethered Arp2/3 complexes we also corrected for drift in the microscope stage using automated tracking of the centers of 2D-Gaussian-fit fluorescence emissions from individual Arp2/3-SNAP649-biotin molecules stably tethered to the microscope slide.

## CoSMoS data analysis

Image processing and kinetic analysis was performed in ImageJ (National Institutes of Health, Bethesda, MD), Origin (OriginLab Corp., Northampton, MA) and with custom programs developed in Matlab (Mathworks, Natick, MA).

### Colocalization dynamics before, during, and after branch formation

To monitor Arp2/3 complex and diVCA colocalization dynamics on the sides of tethered filaments during branch formation (*Figure 2*), we first identified sites of branch formation in image sequences and integrated the fluorescence intensities in a 4 × 4 pixel (0.54 × 0.54 μm) area centered on the branch junctions. Fluorescence intensity traces were created from data in the long- and short-wavelength channels of the microscope, corresponding to Arp2/3-SNAP649 and Cy3-diVCA emissions, respectively. Daughter filament lengths were measured from AF488-actin images by manual tracing the filament contours, and nucleation times were calculated by extrapolating from linear fits of elongation data (*Smith et al., 2013*). Errors in measurement of filament lengths, possibly due to non-uniform incorporation of labeled subunits or fluctuations of filament ends away from the microscope slide (and therefore out of the TIRF excitation field), sometimes led to nucleation times that preceded arrival of Arp2/3 complex on the mother filament (typically ~20% of nucleation events). To ensure that nucleation times were accurate, we restricted our analysis in *Figure 2E* to nucleation events that occurred >0.1 s after Arp2/3 complex appeared on the mother filament side.

To monitor colocalization dynamics of diVCA and filaments on tethered Arp2/3 complexes, we first identified the locations where individual Arp2/3 complexes were stably tethered and integrated the fluorescence intensities at those locations in both emission channels. To specifically analyze diVCA-Arp2/3 interactions when Arp2/3 complex is not bound to filament sides, the Arp2/3-SNAP649-biotin and AF488-actin emission were used to identify periods where each Arp2/3 complex was tethered and no filament colocalized. diVCA binding and dissociation kinetics were then analyzed during those periods at each tethered-Arp2/3 location, as described below. In the rare cases when a mother filament colocalized with tethered-Arp2/3 and nucleation occurred, daughter filament lengths were measured (as above) and the interactions of diVCA with branch junctions were separately analyzed, in periods following nucleation during which tethered-Arp2/3 was continuously observed.

### Binding and dissociation kinetics

To detect times at which diVCA and Arp2/3 complex associated and detached from tethered filament sides, as well as times of association and dissociation of diVCA from tethered Arp2/3 complexes, a combination of automated colocalization detection algorithms (*Crocker and Grier, 1996*; *Friedman and Gelles, 2012*; *Smith et al., 2013*) and manual inspection of images were used. First, the spatial and temporal coordinates of single molecule fluorescence were detected using custom particle tracking software (*Crocker and Grier, 1996*), applied to image sequences that had been filtered by averaging of sequential frames (5-frame sliding average for Arp2/3 complex, 2-frame sliding average for diVCA). These coordinates were then compared to manually selected target sites: either a mask that circumscribed an observed tethered filament or the location of a tethered Arp2/3 complex. Colocalization was scored if the coordinates agreed within ~0.8 μm from the center line of a tethered filament mask or within ~0.3 μm of a tethered-Arp2/3 location. These colocalization records were used in combination with integrated fluorescence intensity traces and image sequences (not subjected to adjacent frame averaging) to guide the ultimate discrimination of bound and dissociated intervals.

Quantification of the kinetics of diVCA-Arp2/3 and Arp2/3-filament interactions was achieved by fitting distributions of bound and dissociated intervals to one-, two-, or three-exponential functions, as described previously (*Smith et al., 2013*). Fits were performed with maximum likelihood algorithms and errors were determined by bootstrapping. The times until first appearance of an Arp2/3 complex

on resolvable (0.4 µm) segments of tethered filaments were fit with a single-exponential function to find the second order rate constants for filament side binding ($k_{A+}$; **Table 1**). The distribution of lifetimes of coincidently bound Arp2/3 complex and diVCA on filament sides were fit with three- and two-exponential functions, respectively (**Figure 2D**; see below for how coincidence was determined). Given the limited number of observations of branch formation events (<70 for each diVCA construct) the dwell times of diVCA on filament-bound Arp2/3 complexes prior to nucleation were not fit to an exponential decay, rather, they were averaged to determine the mean lifetime of diVCA on the nascent branch ($\tau_V^*$; **Figure 6C**). For tethered-Arp2/3 experiments, the second order rate constant for diVCA binding was obtained from single-exponential fits to the intervals where tethered-Arp2/3 was unoccupied by diVCA ($k_{V+}$; **Figure 4—figure supplement 2A–C**), whereas the characteristic lifetimes were obtained from double-exponential fits to the distributions of intervals where tethered-Arp2/3 was occupied by diVCA ($\tau_{V1}$, $\tau_{V2}$; **Figure 4D**). There was no evidence that the biexponential distribution of diVCA-Arp2/3 complexes reflects two distinct forms of Arp2/3 complex that do not interconvert on the timescale of our experiment (~30 min). This was assessed for the V* mutant by calculating the fraction of diVCA-V*-Arp2/3 complexes that lasted <10 s for each tethered Arp2/3 complex observed, which showed a single distribution centered at ~50%, consistent with the ensemble distribution (**Figure 4D**). The effect of photobleaching on the apparent dissociation rate of Cy3-diVCA constructs was assessed and corrected by analyzing the dependence of $\tau_{V2}$ on the excitation laser power (**Figure 4—figure supplement 2D**). Laser power was measured by intercepting the beam path between dichroic mirror D3 and mirror M5 (Figure 1 in **Friedman et al., 2006**). All binding rates and lifetime distributions were corrected for non-specific binding of Arp2/3 complex or diVCA to the microscope slide, as described previously (**Smith et al., 2013**).

## Arp2/3-diVCA coincidence, co-release, and branching efficiencies on filament sides

For each Arp2/3 complex detected on the side of a tethered filament, we evaluated what fraction arrived complexed with diVCA ($f_{AV}$). Coincident arrival of Arp2/3 complex and diVCA was determined if a diVCA appeared within 0.27 µm (2 pixels) from the location and within 0.15 s (3 frames) of the arrival time of the Arp2/3 complex, given conservative estimates of spatial and temporal resolutions of the frame-averaged dual-channel particle tracking methods. Although we cannot distinguish between coincident arrival and sequential arrival of Arp2/3 complex and diVCA within 0.15 s, we would not expect that diVCA could bind so rapidly after formation of Arp2/3-filament complexes, given the measured rate of binding to Arp2/3 complexes off-filament and the diVCA concentration in solution (~$10^8$ M$^{-1}$s$^{-1}$ × 5 nM = ~0.5 s$^{-1}$). We also corrected the total number of Arp2/3 complexes that appeared on filament sides with and without diVCA by the expected number of Arp2/3-filament encounters missed because they were too transient (lifetime <0.1 s) to be detected. This correction factor ($1/p_0$) was calculated using the multi-exponential fits to the Arp2/3-filament lifetime distributions (survival probability p[$t$]), with and without coincident diVCA, evaluated at the minimum detectible event duration: $p_0 = p(0.1\ s)$. This detection efficiency was typically 0.5–0.6. The same correction factor was applied to the second order rate constants of filament side binding, $k_{A+}$.

Similarly, co-release of an Arp2/3 complex and diVCA was determined if the disappearance of diVCA was within 0.15 s from the disappearance of the Arp2/3 complex with which it coincidently bound the filament side. The subset of diVCA-Arp2/3-filament complexes from which diVCA was released leaving Arp2/3 complex bound to the filament, for at least 0.15 s after diVCA release, was counted toward the diVCA release efficiency ($f_{V-}$). Also, the fraction of diVCA-Arp2/3-filament complexes or Arp2/3-filament complexes that nucleated a daughter filament was counted as the branch formation efficiency ($f_B$) with or without diVCA, respectively. The number of binding and release events in each category (Arp2/3 complex alone, coincident with diVCA, Arp2/3-diVCA co-released, and diVCA release before Arp2/3 complex) were corrected by the number of observations in each category on regions of the field of view that did not contain a filament, so as to correct for non-specific interactions with the microscope slide.

Finally, the rate of release of diVCA from the nascent branch was calculated as: $k_{V-}^* = f_{V-}/\tau_V^*$ (**Figure 6D**). This relation results from the fact that the observed lifetime of the nascent branch intermediate is limited by the combination of the dissociation of diVCA from the filament-bound Arp2/3 complex (proceeds with rate $k_{V-}^*$) and the dissociation of diVCA-Arp2/3 complex from the filament

(proceeds with rate $k_{A-}$), such that $\tau_V^* = 1/(k_{A-} + k_{V-}^*)$. The lifetimes (*Figure 6C*) are very similar for all diVCA constructs because the dissociation of diVCA-Arp2/3 complexes from filament sides is dominant ($k_{A-} \gg k_{V-}^*$) and $k_{A-}$ is not significantly affected by VCA mutations. Note that the proposed mechanism (*Figure 6A*) predicts that the nominal lifetime of nascent branches is the same whether we select all nascent branches or only those that terminate in diVCA release and subsequently form branches. The fraction of nascent branches that release diVCA is determined by kinetic competition between the two pathways of nascent branch disassembly: diVCA release leaving an Arp2/3-filament complex (with rate $k_{V-}^*$; labeled green arrow in *Figure 6A*) and filament release leaving a diVCA-Arp2/3 complex (with rate $k_{A-}$; thick red arrow in *Figure 6A*). This yields $f_{V-} = k_{V-}^*/(k_{A-} + k_{V-}^*)$, and thus we obtain $k_{V-}^* = f_{V-}/\tau_V^*$.

We compared the diVCA release rate to the second order rate of overall branch formation calculated from the single molecule data as: $k_B = k_{A+}[f_{AV} f_B(+diVCA) + (1 - f_{AV}) f_B(-diVCA)]$ (*Figure 5A*, *Figure 6E*). The values of all parameters used in these calculations, for all diVCA constructs tested, are indicated in *Table 1*. To calculate the probability that the measured correlation coefficients between diVCA activity towards Arp2/3 complex and the rate of its release from the nascent branch (*Figure 6E* and *Figure 6—figure supplement 1*) arose by chance (i.e., that the two quantities are not in fact correlated), the data for each diVCA construct was resampled (100,000 times) based on the anisotropic 2D-Gaussian distribution defined by the error bars on each measurement. In each sample, activity and release rate values for the four diVCA constructs were randomly permuted, and the correlation coefficient was calculated. The probability p was then calculated as the definite integral of the normalized distribution of the sample correlation coefficients from the correlation coefficient of the experimental data to one. To the extent that replicate measurements may have correlated sources of error, we may overestimate the magnitude of the errors and therefore underestimate p.

Standard errors on parameters determined from fits to distributions of binding times or lifetimes were calculated by bootstrapping. Binding rate constants were reported as means ± S.E. from multiple replicate experiments. Errors on the fractions of Arp2/3 complex observations that appeared coincident with diVCA ($f_{AV}$), those that released diVCA ($f_{V-}$), and those that formed branches ($f_B$), were calculated using counting statistics: $\delta f = \sqrt{[f (1 - f)/N]}$. Errors on quantities calculated from multiple measured parameters (such as $k_B$), were calculated using propagation of errors.

## Acknowledgements

We thank L Friedman and J Chung for assistance with TIRF microscopy and analysis, CA Ydenberg for help with bulk actin assembly kinetics, and members of the Goode lab for help with actin purification and labeling, A Okonechnikov for single-molecule analysis software development, and L Helgeson and B Nolen for sharing the results of unpublished work.

## Additional information

### Competing interests

IRC, M-QX: New England Biolabs manufactures and sells the SNAP-tag system, components of which are used in this study. BLG: Reviewing editor, *eLife*. The other authors declare that no competing interests exist.

### Funding

| Funder | Grant reference number | Author |
| --- | --- | --- |
| National Institutes of Health | GM43369 | Jeff Gelles |
| Howard Hughes Medical Institute | | Michael K Rosen |
| Welch Foundation | I-1544 | Michael K Rosen |
| National Science Foundation | MRSEC 0820492 | Karen Daugherty-Clarke, Bruce L Goode |
| National Institutes of Health | GM56322 | Michael K Rosen |
| National Institutes of Health | GM63691 | Bruce L Goode |
| National Institutes of Health | GM098143 | Jeff Gelles |

The funders had no role in study design, data collection and interpretation, or the decision to submit the work for publication.

## Author contributions

BAS, SBP, Conception and design, Acquisition of data, Analysis and interpretation of data, Drafting or revising the article; LKD, KD-C, M-QX, Acquisition of data, Analysis and interpretation of data; IRC, Acquisition of data, Analysis and interpretation of data, Drafting or revising the article; BLG, Conception and design, Drafting or revising the article; MKR, JG, Conception and design, Analysis and interpretation of data, Drafting or revising the article

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
