## [Decision Letter]

Thank you for sending your work entitled “Tri-color single-molecule imaging shows WASP release from the nascent branch triggers actin nucleation by Arp2/3 complex” for consideration at *eLife*. Your article has been favorably evaluated by a Senior editor and 3 reviewers, one of whom, Wesley Sundquist, is a member of our Board of Reviewing Editors, and one of whom, Thomas Pollard, has agreed to reveal his identity.

The Reviewing editor and the other reviewers discussed their comments before we reached this decision, and the Reviewing editor has assembled the following comments to help you prepare a revised submission.

The authors address how WASP/NPF family members (as represented by the minimal dimerized VCA activation region) act to stimulate actin branching through the Arp2/3 complex. They conclude that: 1) diVCA stimulates initial assembly of the nascent branch by associating with Arp2/3 complex and actin and promoting association of this “nascent branch complex” to the side of the actin filament, and 2) diVCA (but not Arp2/3 complex) then dissociates to create a complex that is elongated into a branch. The latter point is the major new contribution of the manuscript.

In support of the idea that dissociation of the diVCA component triggers branch formation, the authors use three-color fluorescence experiments to demonstrate that: 1) Arp2/3 complex and diVCA (and presumably also actin monomers) bind filaments simultaneously. These nascent branch complexes usually associate non-productively and the complex typically dissociates as a unit. In a small subset of the cases, however, the diVCA component alone dissociates, and this almost invariably leads to branch formation. Furthermore, diVCA release and initiation of branch elongation occur at exactly the same time (within experimental error). The data in Figure 1 appear to be of high quality, and imply that release of diVCA is likely an obligatory step in the commitment to branch formation. 2) diVCA does not bind Arp2/3 complexes once branches have formed (Figure 2), again consistent with the idea that the presence of diVCA is incompatible with branch formation, and that diVCA departure induces a conformational change(s) that destroy the diVCA binding site and stimulates branch formation. 3) Mutations that alter the half life of diVCA within the nascent branch complexes also alter the rate of branching stimulation, and there is a strong positive correlation between the diVCA dissociation rate constant and the rate constant for branch formation (Figures 3 and 4). These observations support a model that explains how membrane-associated WASP complexes can stimulate actin branching, but then dissociate to create space for subsequent actin polymerization (Figure 5).

This work is interesting and important, being the first to document at the single molecule level the timing of the interactions of Arp2/3 complex with the partners required to make an actin filament branch – the nucleation promoting factor VCA, actin filaments and actin monomers. The data are generally of high quality, and enhance our understanding of the mechanism of Arp2/3 complex activation by revealing NPF release as a key step in the activation process. Moreover, the study is a nice example of the application of single-molecule imaging to address an important yet incompletely defined biochemical pathway. The manuscript should therefore be acceptable for publication once the following three important issues have been addressed: Additional points noted below are also important, and should be considered carefully and a response provided.

1) Figure 1: We would urge the authors to provide a more explanatory diagram at the beginning of the paper to draw in the outsider and help explain the points being made in the Introduction, and the key question that is being tested. Figure 1A does this to some extent, but it's small and could be much more explanatory. Figure 1E is very busy and it is difficult to distinguish data points for diVCA wild type from diVCA mutants. The data might be more clearly illustrated in four separate graphs without taking up too much more space. Also, there appear to be outliers for which diVCA release occurs after daughter nucleation. These are not mentioned explicitly in the text, and it is unclear whether these outliers are the ones that fall within “experimental uncertainty”. Moreover, it is unclear what the error bars in this figure represent (and, more generally, the authors need to be careful to describe what error bars represent in all of the different graphs in the figures).

2) Although the data with diVCA wild type is pretty clear, the data with the diVCA mutants is sometimes a bit confusing. For example, the data in Figure 3C suggests that diVCA-C* dissociates more rapidly from Arp2/3 complex than diVCA-wt. However, the data in Figure 4E suggest that the release rate constant is similar for diVCA-C* and diVCA-wt. Might this is due to the difference between free versus filament-bound Arp2/3 complex? Also, the error bars for some of these panels are fairly large, and it isn't clear whether the observed parameters are statistically significantly different. The authors should therefore carry out statistical tests to determine whether there are significant differences in the measured parameters for wild type and mutant diVCA in Figure 4 (and elsewhere). Without that, it is hard to say whether the trend with the mutants fully supports the main conclusions.

3) Please also: A) reorganize the figures, and B) reframe the presentation of the central question under study to reflect previous literature precedents more accurately.

Below, we have raised a series of specific issues for your consideration. We acknowledge that authors have the right to write their own paper, but believe you will agree that the manuscript will be improved once these different issues are addressed, particularly those involving accurate literature citations.

Additional important points to consider:

1) Figure organization

The figures are currently organized based on the methodology rather than on logical flow. The panels in Figure 2 about mutant VCA's beg for the information in Figure 3. We therefore suggest that you either: A) Switch the figure order so that you present the nature of these mutations and fully explore the binding of VCA wild type and mutants before presenting what happens when a mother filament binds these mutant VCA's. Having Figure 2 after Figure 3 will also provide a better opportunity to integrate the two (currently neglected) panels in Figure 2 on mutant VCA's into the presentation of branching by mutant VCA's in Figure 4. B) Alternatively, consider merging Figure 2A,B (diVCAwt) with Figure 1 and merging the remaining panels in Figure 2B (diVCAmutants) with Figure 3.

2) Presentation

Introduction

A fundamental issue is that the study is presented as resolving the literature “paradox” that having WASP tethered to the plasma membrane by its activators (Rho-family GTPases, polyphosphoinositides and SH3 domain proteins) restricts filament growth. In fact, the present work confirms the existing hypothesis about the order of events. That is super, but it's not resolving a “paradox” or “conundrum”.

This concern is best illustrated by a review of the background leading up to this study. In 1999 Egile et al. and Higgs et al. found that the V motif binds the barbed end of actin, so the V-actin complex does not bind the pointed end of actin filaments and inhibits elongation at that end. Egile et al. imagined that VCA was a “motor of insertional polymerization” that would bind actin monomers and deliver them to the end of the filament and then dissociate rapidly. The current work disproves this hypothesis. On the other hand, Higgs concluded: “Since WA (VCA) only slightly inhibits barbed end elongation at the concentrations we tested, it must rapidly dissociate from the filament barbed end upon addition of WA-bound monomer. Profilin has a similar effect on polymerization; it inhibits nucleation and pointed end elongation but not barbed end elongation.” Thus the concept that rapid V dissociation is required for barbed end elongation is 14 years old.

Next, Marchand et al. (2001) showed directly that VCA regions of WASP family proteins bind to and dissociate from actin and Arp2/3 complex rapidly. They measured koff from actin monomers to be 3/s and from Arp2/3 complex to be 0.2/s.

Dayel (2001) discovered that hydrolysis of ATP bound to Arps promotes dissociation of VCA from Arp2/3 complex during branch formation. They concluded “binding N-WASP brings the actin monomer attached to the WH2 domain of N-WASP in contact with the Arp2/3 complex and this stimulates ATP hydrolysis. Step 2: Hydrolyzing ATP to ADP-Pi causes a conformational change on the complex, forming a stable nucleus among Arp3, Arp2, and the conventional actin monomer. Step 3: A new actin filament polymerizes from this nucleus. Step 4: Phosphate release from Arp2/3 complex decreases the affinity for N-WASP and allows the Arp2/3 complex to release its membrane-associated activator.” Dayel (2004) looked at the timing more carefully and concluded “it is very likely that ATP hydrolysis on Arp2, like actin, provides a timing signal to the system. ATP hydrolysis on Arp2/3 would promote release of VCA from the complex and allow a new actin branch to move away from the site of its creation (Dayel et al. 2001).” The enclosed drawing from Dayel (2004) summarizes their understanding of the process. This drawing is very similar to Figure 5 in the current paper. Neither Dayel paper is cited, and this seems inappropriate. [See attached PDF version for image of Figure 5B from Dayel (2004).]

A new paper from the Nolen lab (Hetrick et al, Chem Bio, 2013) has an updated version of the Dayel diagram in their Figure 6. Of course the present authors would not have known about this new paper, but it illustrates that the basic idea is central to the thinking in the field.

Although it is less well established that the C motif binds the barbed end of Arp3 and possibly Arp2, previous workers postulated that C might inhibit binding of the first two subunits of the daughter filament. Therefore, several investigators emphasized that early in the nucleation process VCA must inevitably dissociate from Arp2/3 complex and the rates of dissociation measured previously mean that the delay would be on the order seconds. A recent paper from the current authors (Smith et al.) reported the important insight that most interactions of Arp2/3 complex with the side of an actin filament are transient, explaining why Arp2/3 complexes associate stably with actin filaments so slowly (Beltzer; Ti).

An account of the background with these points would be an appropriate place to begin this paper. Rather than investigating a paradox, this work tests the main hypothesis in the field, which has been put forward by multiple labs.

Other aspects of the Introduction are a nice account of branching by Arp2/3 complex but several of the citations are misplaced. See the attached pdf with notes.

Results

It might be noted that the observation that most Arp2/3 complexes binding to filaments are associated with di-VCA is exactly what is expected, given the thermodynamic argument in Ti et al. (2011), namely that bound VCA increases the affinity of Arp2/3 complex for actin filaments about 30 fold.

The conclusion “daughter nucleation is essentially always accompanied by Arp2/3 complex retention and diVCA release” is exactly what is expected from previous work, although it is certainly valuable to confirm this order of events by these lovely single molecule observations. Congratulations on the lovely experiments.

“Thus, diVCA release may serve as the trigger for daughter nucleation.” The wording of this conclusion reinvents the concept of the actin nucleus, which according to historical precedent is the smallest actin oligomer that elongates like a filament. Classic simulations of the time course of spontaneous actin polymerization (labs of Frieden, Pollard and Korn) showed that the nucleus is a trimer. Sept confirmed this interpretation with Brownian dynamics simulations. Even though the trimer is unstable, it adds subunits like any longer filament. The nucleus in the case of Arp2/3 complex is therefore likely consists of the two Arps and one or two actin subunits. Therefore, the nucleus is formed when the first and second actin subunits bind Arp2/3 complex, which can surely happen prior to dissociation of V from the two actins. Thus what is observed in this experiment is not nucleation, but the beginning of elongation, which appears to be limited by dissociation of the diVCA as suggested by others. Therefore throughout the paper “nucleation” should be replaced with “elongation” of the daughter filament (which is what is observed).

“Most of the tethered Arp2/3 complexes (>80 %) were observed to bind diVCA.” Please explain. Does this mean that 20% never bound di-VCA? Were they dead?

Machesky et al. (PNAS, 1999) (not cited) showed that mother filaments are required for nucleation, not Mullins et al. (1998).

The authors should check carefully whether or not there is a precedent for their finding that “the affinity of diVCA for isolated Arp2/3 complex is high, whereas the diVCA affinity for Arp2/3 complex in the branch junction is comparatively low.” Regardless, that observation makes sense given that both the C and V binding sites are likely to be occluded in the branch junction, and since the A motif binds weakly to Arp2/3 complex (Kd ∼ 9 µM, Marchand, 2001).

The text states “The goal was to modestly perturb VCA interactions with Arp2/3 complex or actin without altering the mechanism by which diVCA stimulates branch formation.” This statement reveals a misunderstanding about the thermodynamics and should be rephrased. It is highly unlikely that one could change the affinity of VCA for its receptors without altering the activation process. The reason is that activation depends on the free energy from Arp2/3 complex binding its three ligands (VCA, actin monomers and actin filament), so if one perturbs any of these free energy changes, one inevitably perturbs activation.

“With the A* mutant dissociating from Arp2/3 complex more slowly than wild type.” Were you surprised? Why should this be true?

“Both wild type and mutant complexes displayed lifetime distributions that were fit well with two exponential components (Figure 3D, Figure 3–figure supplement 2D, Table 1), indicating that at least two distinct diVCA-Arp2/3 complex assemblies could form.” The semi-log plot makes it difficult to see the fast component in some of these plots. What is the behavior of each individual Arp2/3 complex? Were they consistently in one of these states or did one Arp2/3 complex behave differently over time? This behavior deserves a more detailed analysis and presentation.

The presence of multiple complexes is expected – really? Alternatively, given the bivalent nature of the ligand and the receptor, loss of function of one site on a subset of ligands or receptors could result in the observed two lifetimes. Can you rule out this explanation?

Figure 4A (and the associated supplemental figure) show that wild type diVCA boosted Arp2/3 complex-dependent polymerization only 1.5 fold (microscopy) to 2.0 fold (bulk biochemistry). This result agrees with previous work from the Rubenstein lab on bulk samples (not cited), but it deserves some comment. Describe what was observed by microscopy about branches formed by Arp2/3 complex in the absence of VCA, since this unusual feature of budding yeast Arp2/3 complex is poorly understood. The experiments in the magenta and green boxes in Figure 2 also need more detailed explanations.

Discussion

Arasada (2011) has another estimate of NPF density that is more relevant this work on budding yeast Arp2/3 complex than the other examples, namely between 16,000 and 45,000 NPF's per µm2 at sites of endocytosis where actin polymerization depends on Arp2/3 complex.

Only Ti et al. (2011) presented a molecular structure of Arp2/3 complex showing how the C-region of VCA may occupy a similar location on Arp3 that the V-region occupies on actin. The other cited work infers the location of the C binding sites from indirect data.

---

## [Author Response]

*1)*
*Figure 1**: We would urge the authors to provide a more explanatory diagram at the beginning of the paper to draw in the outsider and help explain the points being made in the Introduction, and the key question that is being tested.*
*Figure 1A*
*does this to some extent, but it's small and could be much more explanatory*.

We have added a new figure, which we cite in the Introduction. It is intended to illustrate the biological context and to define the question being addressed in the research.

*Figure 1E*
*is very busy and it is difficult to distinguish data points for diVCA wild type from diVCA mutants. The data might be more clearly illustrated in four separate graphs without taking up too much more space*.

We have replaced Figure 1E with a clearer panel (now Figure 2) that shows the wild-type data only. Presentation of the mutant data is deferred until later in the paper. We think it important to show all mutant and wild type data superimposed (so that readers can see that the distribution of points is similar in all four cases), but we agree with the reviewers that such a presentation makes it difficult to evaluate which points are significant outliers. We therefore compromised by presenting both the original panel (now Figure 5) and new panels that present the data for the four constructs individually (in Figure 2 and in Figure 5—figure supplement 3).

*Also, there appear to be outliers for which diVCA release occurs after daughter nucleation. These are not mentioned explicitly in the text, and it is unclear whether these outliers are the ones that fall within “experimental uncertainty”*.

In the wild type data (Figure 2) there are no significant outliers as can be seen by the fact all points are on or above the diagonal dashed line, or have error bars that cross it. In the mutant data (Figure 5—figure supplement 3) there is one point for V*, zero for C*, and one for A* that could be significant outliers, out of 41, 49, and 27 observations, respectively. We have marked these two points with asterisks and now explicitly discuss them in the figure caption.

*Moreover, it is unclear what the error bars in this figure represent (and, more generally, the authors need to be careful to describe what error bars represent in all of the different graphs in the figures)*.

Error bars in this figure represent S.E. of the daughter nucleation time measurements derived from fits like those shown in Figure 2, blue. This is now stated in the figure caption and described in the methods. We have reviewed all figure captions and specified the meaning of the error bars where this was not done before.

*2) Although the data with diVCA wild type is pretty clear, the data with the diVCA mutants is sometimes a bit confusing. For example, the data in*
*Figure 3C*
*suggests that diVCA-C* dissociates more rapidly from Arp2/3 complex than diVCA-wt. However, the data in*
*Figure 4E*
*suggest that the release rate constant is similar for diVCA-C* and diVCA-wt. Might this is due to the difference between free versus filament-bound Arp2/3 complex*?

Yes, that is the likely explanation. Indeed, the reason for the measurements in Figure 4E (it is now Figure 6) is to characterize the dissociation kinetics of VCA from filament–bound Arp2/3 complex. They do in fact turn out to be different from those with off-filament Arp2/3 complex. The text now states this clearly.

*Also, the error bars for some of these panels are fairly large, and it isn't clear whether the observed parameters are statistically significantly different. The authors should therefore carry out statistical tests to determine whether there are significant differences in the measured parameters for wild type and mutant diVCA in*
*Figure 4*
*(and elsewhere). Without that, it is hard to say whether the trend with the mutants fully supports the main conclusions*.

There are two kinds of conclusions related to comparison of the wild type and mutant constructs (data that are now in Figure 6 as well as in some of the figure supplements) that require statistical support. The first are pairwise comparisons (e.g., statements that a particular mutant is different than wild type with respect to a particular measurement). For each of the pairwise comparisons that we utilize, we have now performed a significance test and p values are given in the text or figure captions. The second type of conclusion is the apparent correlations (Figure 6 and Figure 6—figure supplement 1). For each of these we now report the correlation coefficients and further report a bootstrap calculation, which shows that it is unlikely (p = ∼0.005 and ∼0.03, respectively) that the high measured correlation coefficient was obtained by chance from two quantities that are in truth uncorrelated. This analysis is described in the Materials and methods section of the revised manuscript.

*3) Please also: A) reorganize the figures, and B) reframe the presentation of the central question under study to reflect previous literature precedents more accurately. Below, we have raised a series of specific issues for your consideration. We acknowledge that authors have the right to write their own paper, but believe you will agree that the manuscript will be improved once these different issues are addressed, particularly those involving accurate literature citations*.

As described below, we have largely incorporated these suggestions. In a few cases we did not agree that the suggested changes would be improvements, and we explain why below.

*Additional important points to consider*:

1) Figure organization

*The figures are currently organized based on the methodology rather than on logical flow. The panels in Figure 2 about mutant VCA's beg for the information in Figure 3. We therefore suggest that you either: A) Switch the figure order so that you present the nature of these mutations and fully explore the binding of VCA wt and mutants before presenting what happens when a mother filament binds these mutant VCA's. Having Figure 2 after Figure 3 will also provide a better opportunity to integrate the two (currently neglected) panels in Figure 2 on mutant VCA's into the presentation of branching by mutant VCA's in Figure 4. B) Alternatively, consider merging Figure 2A,B (diVCAwt) with Figure 1 and merging the remaining panels in Figure 2B (diVCAmutants) with Figure 3*.

The principal concern seems to be that we originally had the data from the mutants and the wild type presented together in figures in a way that was inconsistent with the order of presentation in the text. We have now split out the mutant data so that all data are presented in the order described in the text. This is essentially adopting the order of presentation described above as alternative B, although we have grouped the panels slightly differently than what was suggested.

Specifically, we have:

• Added the new Figure 1 (as described in response to referee comment 1).

• Replaced Figure 1E (now Figure 2) with version showing wild type data only.

• Replaced Figure 2 (now Figure 3) with version showing wild type data only.

• Figure 3 (now Figure 4) is unchanged except for minor edits.

• Added Figure 5 consisting of the mutant data removed from the earlier figures (old 4A, old 1E and old 2B).

• Old Figure 4B–F is now Figure 6.

• Old Figure 5 (now Figure 7) is edited to clarify but otherwise unchanged.

2) Presentation

Introduction

*A fundamental issue is that the study is presented as resolving the literature “paradox” that having WASP tethered to the plasma membrane by its activators (Rho-family GTPases, polyphosphoinositides and SH3 domain proteins) restricts filament growth. In fact, the present work confirms the existing hypothesis about the order of events. That is super, but it's not resolving a “paradox” or “conundrum”*.

We did not intend to present our work as resolving a “literature paradox”. Rather, we are addressing the inherent conflict between the need for Arp2/3 complex to associate with membrane bound activators at one stage of the daughter filament formation process and to be distant from those activators at a subsequent stage. By directly observing the order of events by which the complex moves from one stage to the next, our study defines the mechanism by which this conflict is resolved.

In order to more clearly define the problem that is resolved by our study, and to more fully place the work in the context of the existing literature, we have slightly rephrased the Abstract and made substantial changes to the Introduction. The changes to the Introduction include:

• We added Figure 1 and accompanying text in the Introduction. We feel that these work together to more clearly define the most important problem that is addressed by our study.

• We have expanded and reorganized the beginning of the Introduction to include additional literature citations (mostly ones suggested below by the reviewers) to more thoroughly place the work in the context of existing knowledge about how VCA proteins function to stimulate daughter filament initiation.

• We added a new paragraph that emphasizes (with numerous literature citations) that separation of the new branch junction from the membrane- or surface-bound activator is already well established in a variety of biological contexts and in vitro.

• We added a new penultimate paragraph to the Introduction that explicitly states (with citations) that VCA departure prior to initiation of daughter filament growth was previously hypothesized.

• We removed the words “paradox” and “conundrum” in an effort to avoid any implication that there are inconsistencies between different results in the existing literature, an implication that we did not intend.

We believe that these changes will largely address the concerns of the referees about the original introduction. We anticipate that they will also make the context of the work clearer to readers, particularly those who may be less familiar with prior work in the field.

*This concern is best illustrated by a review of the background leading up to this study…*.

Although we concur with most of what is said below, there are a few stated conclusions and implications that we do not think are well supported by the literature or that are contradicted by it. However, since it is explicit that some of what follows is intended to be illustrative, we have responded only to parts that explicitly recommend changes to the manuscript.

*…In 1999 Egile et al. and Higgs et al. found that the V motif binds the barbed end of actin, so the V-actin complex does not bind the pointed end of actin filaments and inhibits elongation at that end. Egile et al. imagined that VCA was a “motor of insertional polymerization” that would bind actin monomers and deliver them to the end of the filament and then dissociate rapidly. The current work disproves this hypothesis. On the other hand, Higgs concluded: “Since WA (VCA) only slightly inhibits barbed end elongation at the concentrations we tested, it must rapidly dissociate from the filament barbed end upon addition of WA-bound monomer. Profilin has a similar effect on polymerization; it inhibits nucleation and pointed end elongation but not barbed end elongation.” Thus the concept that rapid V dissociation is required for barbed end elongation is 14 years old*.

*Next, Marchand et al. (2001) showed directly that VCA regions of WASP family proteins bind to and dissociate from actin and Arp2/3 complex rapidly. They measured koff from actin monomers to be 3/s and from Arp2/3 complex to be 0.2/s*.

*Dayel (2001) discovered that hydrolysis of ATP bound to Arps promotes dissociation of VCA from Arp2/3 complex during branch formation. They concluded “binding N-WASP brings the actin monomer attached to the WH2 domain of N-WASP in contact with the Arp2/3 complex and this stimulates ATP hydrolysis. Step 2: Hydrolyzing ATP to ADP-Pi causes a conformational change on the complex, forming a stable nucleus among Arp3, Arp2, and the conventional actin monomer. Step 3: A new actin filament polymerizes from this nucleus. Step 4: Phosphate release from Arp2/3 complex decreases the affinity for N-WASP and allows the Arp2/3 complex to release its membrane-associated activator.” Dayel (2004) looked at the timing more carefully and concluded “it is very likely that ATP hydrolysis on Arp2, like actin, provides a timing signal to the system. ATP hydrolysis on Arp2/3 would promote release of VCA from the complex and allow a new actin branch to move away from the site of its creation (Dayel et al. 2001).” The enclosed drawing from Dayel (2004) summarizes their understanding of the process. This drawing is very similar to Figure 5 in the current paper. Neither Dayel paper is cited, and this seems inappropriate. [See attached PDF version for image of Figure 5B from Dayel (2004).*]

We have added a citation to both Dayel papers in the new section in the final paragraph of the Introduction. As an aside, we note here that the image provided by the referees shows a hypothesis that is not identical to our conclusions. First, it posits a key role for ATP hydrolysis by Arp2/3 complex, a question about which our work is silent but which is addressed in subsequent studies (Martin, 2006; Ingerman, 2013). (These subsequent studies suggest that hydrolysis does not, in fact, accompany initiation of the daughter filament, as suggested in Dayel (2004), but rather is related to debranching.) Second, it does not address the central concern of our paper, which is the order of VCA release and initiation of daughter filament polymerization. It shows the two lumped into a single, final step so that their order is not specified. In this respect the figure is more akin to our Figure 1 (which illustrates the key question we are asking in our study) than it is to our Figure 7 (which illustrates the answer).

*A new paper from the Nolen lab (Hetrick et al, Chem Bio, 2013) has an updated version of the Dayel diagram in their Figure 6. Of course the present authors would not have known about this new paper, but it illustrates that the basic idea is central to the thinking in the field*.

We have added citations to the Hetrick paper at the relevant locations in the Introduction. We would respectfully disagree with the reviewers’ implication that the inclusion of this model in a single paper published a few months ago establishes that the idea is central to the thinking in the field. In our estimation, a more reasonable conclusion is that some features in the model published by Hetrick et al. were influenced by the as-yet unpublished work from the Nolen lab (we cite their submitted publication) and perhaps also by reports of our work reported in this paper, which has been presented at conferences.

*Although it is less well established that the C motif binds the barbed end of Arp3 and possibly Arp2, previous workers postulated that C might inhibit binding of the first two subunits of the daughter filament. Therefore, several investigators emphasized that early in the nucleation process VCA must inevitably dissociate from Arp2/3 complex and the rates of dissociation measured previously mean that the delay would be on the order seconds*.

While there is a great deal of literature that touches on related points, we were unable to find any published studies that explicitly “postulated that C might inhibit binding of the first two subunits of the daughter filament”.

*A recent paper from the current authors (Smith et al.) reported the important insight that most interactions of Arp2/3 complex with the side of an actin filament are transient, explaining why Arp2/3 complexes associate stably with actin filaments so slowly (Beltzer; Ti)*.

*An account of the background with these points would be an appropriate place to begin this paper. Rather than investigating a paradox, this work tests the main hypothesis in the field, which has been put forward by multiple labs*.

As noted earlier, we have expanded the Introduction to incorporate additional background, including many of the references cited by the referees that we did not already include. We did not include all such material because we regard some of it as related but not essential, and we think that too much additional length in the Introduction would weaken the presentation. Also noted earlier is that we have added a section explaining that it is well established that VCA must depart before the daughter filament elongates too far. We do not agree (for reasons that are in part explained above) that the field has coalesced on a single hypothesized mechanism to explain this. For this reason, we did not (as the reviewers seem to be suggesting) adopt this hypothesis as a rhetorical device to introduce the question being asked in the paper. Instead, we used the new Figure 1 and related discussion. We think this is a more effective presentation and will also make the paper more accessible to readers outside of the field. We realize that the reviewers may not agree with this decision, but we ask that some deference be given to authors’ choices on how they introduce their work.

*Other aspects of the introduction are a nice account of branching by Arp2/3 complex but several of the citations are misplaced. See the attached pdf with notes*.

Thank you for the notes on the pdf. These suggestions have been incorporated by changing the references and/or citing text.

Results

*It might be noted that the observation that most Arp2/3 complexes binding to filaments are associated with di-VCA is exactly what is expected, given the thermodynamic argument in Ti et al. (2011), namely that bound VCA increases the affinity of Arp2/3 complex for actin filaments about 30 fold*.

The observation that nearly all Arp2/3 complexes binding to filaments are accompanied by diVCA is not a reflection of the stability of the ternary complex. Rather, it is mainly a consequence of the fact that Arp2/3 complexes free in solution (i.e., not attached to a filament) are nearly saturated with diVCA under the conditions of the experiments.

*The conclusion “daughter nucleation is essentially always accompanied by Arp2/3 complex retention and diVCA release” is exactly what is expected from previous work, although it is certainly valuable to confirm this order of events by these lovely single molecule observations. Congratulations on the lovely experiments*.

This comment seems to imply that expectations and hypotheses do not satisfy the need for direct experimental data. We agree.

*“Thus, diVCA release may serve as the trigger for daughter nucleation.” The wording of this conclusion reinvents the concept of the actin nucleus, which according to historical precedent is the smallest actin oligomer that elongates like a filament. Classic simulations of the time course of spontaneous actin polymerization (labs of Frieden, Pollard and Korn) showed that the nucleus is a trimer. Sept confirmed this interpretation with Brownian dynamics simulations. Even though the trimer is unstable, it adds subunits like any longer filament. The nucleus in the case Arp2/3 complex is therefore likely consists of the two Arps and one or two actin subunits. Therefore, the nucleus is formed when the first and second actin subunits bind Arp2/3 complex, which can surely happen prior to dissociation of V from the two actins. Thus what is observed in this experiment is not nucleation, but the beginning of elongation, which appears to be limited by dissociation of the diVCA as suggested by others. Therefore throughout the paper “nucleation” should be replaced with “elongation” of the daughter filament (which is what is observed)*.

In the original manuscript, we used the term “nucleation” to refer both to the overall process shown in the final figure and to the specific step in which the daughter filament begins to grow. To minimize any possible confusion, we now refer to the latter as “initiation of daughter filament growth”. We think this terminology is consistent with the point made by the reviewers but is more apt than “elongation”.

*“Most of the tethered Arp2/3 complexes (>80 %) were observed to bind diVCA.” Please explain. Does this mean that 20% never bound di-VCA? Were they dead*?

Yes, <20% never bound di-VCA. There are multiple possible explanations for the apparently inactive fraction, including 1) incompletely assembled “dead” Arp2/3 complexes, 2) fully assembled and active complexes bound to the surface in an orientation or local environment that precludes VCA binding, and 3) presence of a small number of the dye-biotin adducts used to label the Arp2/3 complex that never were or are no longer attached to protein.

*Machesky et al. (PNAS, 1999) (not cited) showed that mother filaments are required for nucleation, not Mullins et al. (1998)*.

Thanks; we now cite the correct reference.

*The authors should check carefully whether or not there is a precedent for their finding that “the affinity of diVCA for isolated Arp2/3 complex is high, whereas the diVCA affinity for Arp2/3 complex in the branch junction is comparatively low.” Regardless, that observation makes sense given that both the C and V binding sites are likely to be occluded in the branch junction, and since the A motif binds weakly to Arp2/3 complex (Kd ∼ 9 µM, Marchand, 2001)*.

We now say that the results are consistent with data from previous studies and cite two. We agree that the result makes sense given the Marchand result and we now point this out in the Discussion.

*The text states “The goal was to modestly perturb VCA interactions with Arp2/3 complex or actin without altering the mechanism by which diVCA stimulates branch formation.” This statement reveals a misunderstanding about the thermodynamics and should be rephrased. It is highly unlikely that one could change the affinity of VCA for its receptors without altering the activation process. The reason is that activation depends on the free energy from Arp2/3 complex binding its three ligands (VCA, actin monomers and actin filament), so if one perturbs any of these free energy changes, one inevitably perturbs activation*.

We disagree that the statement reveals a misunderstanding about the thermodynamics. Rather, it arises from different meanings attached to the term “mechanism” by the authors and the reviewers. To lessen the possibility of being misunderstood, we have substituted the term “reaction pathway” for “mechanism” to clarify that we are making a statement about the sequence of chemical species in the reaction rather than about free energy changes or rate constants of the reaction steps.

*“With the A* mutant dissociating from Arp2/3 complex more slowly than wild type.” Were you surprised? Why should this be true*?

The A* mutation was indeed designed with this goal in mind. In 2001, Zalevsky et al. reported that VCA peptides have an activation step following assembly of VCA, actin and Arp2/3 complex onto the mother filament. The apparent activation rate constant for this step differed between N-WASP, WASP and WAVE1. The origin of this difference was concluded to be in the length of the acidic region, with a longer acidic region (e.g. in N-WASP) giving faster rate and a shorter region (e.g. in WAVE1) giving slower rate. We wondered if the ‘activation step’ might be departure of VCA. If so, shortening the N-WASP acidic region might make it more WAVE1-like, slowing dissociation. We proceeded to essentially reverse the series of experiments used by Zalevsky et al. by deleting several acidic residues from the N-WASP acidic region. Multiple VCA modifications along these lines generated VCA dimers with lower activity in pyrene actin assays than our wild type diVCA. The “A*” mutant had the smallest perturbation (in terms of amino acid changes) and was therefore chosen for further analysis in the single molecule measurements. In the interest of keeping the manuscript focused, we have not included a description of the reasoning involved in this aspect of the experimental design.

*“Both wild type and mutant complexes displayed lifetime distributions that were fit well with two exponential components (Figure 3D, Figure 3–figure supplement 2D, Table 1), indicating that at least two distinct diVCA-Arp2/3 complex assemblies could form.” The semi-log plot makes it difficult to see the fast component in some of these plots. What is the behavior of each individual Arp2/3 complex? Were they consistently in one of these states or did one Arp2/3 complex behave differently over time? This behavior deserves a more detailed analysis and presentation*.

We added a magnified inset so that the fast component of the distributions in Figure 3D (now 4D) is more visible. Individual Arp2/3 complexes show both short and long events. In a more detailed analysis with the V* mutant, which is the one for which the short and long components are most clearly resolved, we saw no evidence that the short and long lifetime components segregated into different subpopulations of individual Arp2/3 complexes. Both results are now stated in the Results section and further details are given in Methods. We agree with the reviewer that the multicomponent lifetime distributions are intriguing: they could potentially shed light on the different modes of interaction between VCA dimers and Arp2/3. However, such an investigation would likely require analysis of Arp2/3 complex mutations and of a more extensive panel of VCA mutations. Such work is beyond the scope of the present study.

*The presence of multiple complexes is expected - really? Alternatively, given the bivalent nature of the ligand and the receptor, loss of function of one site on a subset of ligands or receptors could result in the observed two lifetimes. Can you rule out this explanation*?

We changed this passage to say that the result is “consistent” with the results of the previous studies cited, not “expected” from them. We cannot rule out the alternative explanation raised by the reviewers, but we see no reason to explicitly discuss it in the manuscript since we already give multiple examples of other possible explanations for the observed behavior and there is currently no basis to choose between them. Distinguishing between these alternatives is not essential to the conclusions of our paper.

*Figure 4A (and the associated supplemental figure) show that wild type diVCA boosted Arp2/3 complex-dependent polymerization only 1.5 fold (microscopy) to 2.0 fold (bulk biochemistry). This result agrees with previous work from the Rubenstein lab on bulk samples (not cited), but it deserves some comment*.

We surmise that the reviewers feel that 1.5–2.0 fold activation is somehow anomalously low and, further, that this value is explained by the significant activity of budding yeast Arp2/3 complex in the absence of VCA. However, another factor is that the N-WASP diVCA construct we use was deliberately chosen because it is only a weak activator (this helped us to engineer mutants that increased activation). This is illustrated by data in Figure 2—figure supplement 2 which show for example, that N-WASP VVCA dimers are much better activators than N-WASP VCA dimers like the one we used as our wild type construct. We now mention both factors in the caption to that figure and we cite Wen & Rubenstein (2005) (although they observed high basal activity only with yeast actin and did not see it with rabbit muscle actin).

*Describe what was observed by microscopy about branches formed by Arp2/3 complex in the absence of VCA, since this unusual feature of budding yeast Arp2/3 complex is poorly understood. The experiments in the magenta and green boxes in Figure 2 also need more detailed explanations*.

A thorough presentation of the microscopy observations without VCA was included in our previous publication (Smith et al., PNAS 2013), which we cite. We see no reason to recapitulate that in the present paper, particularly since the small subpopulation of Arp2/3 complexes that bind to filament unaccompanied by detectable di-VCA fluorescence are excluded from our analyses (e.g., in Figure 6). As described above, the data in the magenta and green boxes from Figure 2B have now been moved to Figure 5; they are described in the caption.

Discussion

*Arasada (2011) has another estimate of NPF density that is more relevant this work on budding yeast Arp2/3 complex than the other examples, namely between 16,000 and 45,000 NPF's per µm2 at sites of endocytosis where actin polymerization depends on Arp2/3 complex*.

The densities mentioned in Arasada (2011) appear to be densities of SH3 domains of F-BAR proteins, inferred from the electron microscopy work of Frost et al. Given the similar numbers of Wsp1 in close proximity to the F-BAR proteins, this data does (albeit somewhat indirectly) support the point being made in the manuscript, and we now cite it.

*Only Ti et al. (2011) presented a molecular structure of Arp2/3 complex showing how the C-region of VCA may occupy a similar location on Arp3 that the V-region occupies on actin. The other cited work infers the location of the C binding sites from indirect data*.

We have added citation to the Ti paper. We think that the other references are also appropriate to support the statement made in the manuscript.